*Resource*

# Small RNA interactome of pathogenic *E. coli* revealed through crosslinking of RNase E

Shafagh A Waters[1], Sean P McAteer[2], Grzegorz Kudla[3], Ignatius Pang[1,4], Nandan P Deshpande[1,4], Timothy G Amos[1], Kai Wen Leong[5], Marc R Wilkins[1,4], Richard Strugnell[5], David L Gally[2], David Tollervey[6,**] & Jai J Tree[1,*]

## Abstract

RNA sequencing studies have identified hundreds of non-coding RNAs in bacteria, including regulatory small RNA (sRNA). However, our understanding of sRNA function has lagged behind their identification due to a lack of tools for the high-throughput analysis of RNA–RNA interactions in bacteria. Here we demonstrate that *in vivo* sRNA–mRNA duplexes can be recovered using UV-crosslinking, ligation and sequencing of hybrids (CLASH). Many sRNAs recruit the endoribonuclease, RNase E, to facilitate processing of mRNAs. We were able to recover base-paired sRNA–mRNA duplexes in association with RNase E, allowing proximity-dependent ligation and sequencing of cognate sRNA–mRNA pairs as chimeric reads. We verified that this approach captures *bona fide* sRNA–mRNA interactions. Clustering analyses identified novel sRNA seed regions and sets of potentially co-regulated target mRNAs. We identified multiple mRNA targets for the pathotype-specific sRNA Esr41, which was shown to regulate colicin sensitivity and iron transport in *E. coli*. Numerous sRNA interactions were also identified with non-coding RNAs, including sRNAs and tRNAs, demonstrating the high complexity of the sRNA interactome.

**Keywords** CLIP-Seq; CRAC; EHEC; enterohaemorrhagic *E. coli*; non-coding RNA

**Subject Categories** Methods & Resources; Microbiology, Virology & Host Pathogen Interaction; RNA Biology

**The EMBO Journal (2017) 36: 374–387**

See also: **J Hör & J Vogel** (February 2017)

## Introduction

Advances in RNA sequencing technologies and associated applications have driven a revolution in our understanding of the complexity of the transcriptome. For diverse bacterial species, a single RNA-Seq experiment can reveal hundreds of novel non-coding RNAs. Bacterial small RNA (sRNA) species regulate translation of mRNAs involved in a diverse range of physiological processes including carbon, amino acid and metal ion utilization (Papenfort & Vogel, 2014), horizontal transfer of DNA (Papenfort *et al*, 2015), biofilm formation (Holmqvist *et al*, 2010) and virulence gene expression (Chao & Vogel, 2010). Canonically, sRNAs repress mRNA translation by base pairing that covers the ribosome-binding site and/or directing the transcript for cleavage and degradation. It is now apparent that there are many variations on this canonical theme including activation of translation (Soper *et al*, 2010), repression by cleavage alone (Pfeiffer *et al*, 2009), cleavage inhibition (Papenfort *et al*, 2013), transcriptional attenuation (Bossi *et al*, 2012) and sRNA sponging (Figueroa-Bossi *et al*, 2009; Tree *et al*, 2014; Miyakoshi *et al*, 2015). The majority of sRNAs in *E. coli* require the RNA chaperone Hfq to anneal with target mRNAs (Gottesman & Storz 2011). Hfq can present sRNAs for interaction with the pool of mRNA targets, increasing the local concentration of interaction partners and providing a positively charged lateral surface to aid annealing (Panja *et al*, 2013).

In principal, targets for sRNA interactions can be predicted using sequence-based analysis; however, few sequence or structural features are conserved between the many different sRNA targets, making false positives a major problem (Backofen *et al*, 2014; Künne *et al* 2014). To overcome this, target prediction programmes have used the presence of a tract of 6 or more consecutive base pairs (the seed sequence) and the predicted accessibility of the seed region (Peer & Margalit, 2011). Phylogenetic conservation of seed sequences also improves the likelihood of identifying functionally significant interactions but is not applicable to transcripts encoded

1   School of Biotechnology and Biomolecular Sciences, University of New South Wales, Sydney, NSW, Australia
2   The Roslin Institute, University of Edinburgh, Edinburgh, UK
3   MRC Human Genetic Unit, University of Edinburgh, Edinburgh, UK
4   Systems Biology Initiative, University of New South Wales, Sydney, NSW, Australia
5   Peter Doherty Institute, University of Melbourne, Melbourne, Victoria, Australia
6   Wellcome Trust Centre for Cell Biology, University of Edinburgh, Edinburgh, UK
   *Corresponding author. Tel: +61 2 9385 9142; E-mail: j.tree@unsw.edu.au
   **Corresponding author. Tel: +44 131 650 7005; E-mail: d.tollervey@ed.ac.uk

within variable regions of the genome, such as pathogenicity islands. In consequence, determining the targets for sRNAs and their regulatory function has generally required the investigation of individual RNAs, often by using transcriptomics to indirectly identify mRNAs with altered stability following sRNA expression or depletion.

A number of recent studies have implemented *in vitro* and *in vivo* techniques to directly identify interactions between non-coding RNAs and their RNA targets. These have included approaches using individual microRNAs or bacterial sRNAs as baits, with or without chemical modifications to improve capture of interacting RNAs. High-throughput sequencing allows identification of target RNAs interacting with the bait RNA (Imig *et al*, 2015). This approach unexpectedly identified a spacer region from the tRNA-Leu precursor as a target for RyhB (Lalaouna *et al*, 2015). An approach to experimentally profile transcriptome-wide RNA–RNA interactions in eukaryotic cells has been described that uses proximity-dependent ligation of duplexed RNAs to capture RNA interactions *in vivo* and has been termed CLASH (UV-crosslinking, ligation and sequencing of hybrids) (Helwak *et al*, 2013) (Fig 1A). RNA–RNA duplexes are UV-crosslinked to a protein "bait" allowing selective capture of RNAs and stringent purification of the RNA–protein complex. A small fraction of RNAs covalently bound to the protein remain duplexed during purification and these can be ligated into a single contiguous RNA molecule with T4 RNA ligase (Helwak *et al*, 2013) or by endogenous RNA ligases (Grosswendt *et al*, 2014). An alternative methodology uses a joining linker to ligate the constrained duplex ends of the RNAs (Sugimoto *et al*, 2015). In each case, a proportion of sequencing reads recovered (typically ~1–2%) consist of read segments that non-contiguously map to the transcriptome. These hybrid reads can be identified *in silico* and indicate sites of intra- or intermolecular RNA–RNA interactions occurring on the bait protein.

RNase E is an endonuclease that plays key roles in both the catalytic activity and assembly of the RNA degradosome, a complex responsible for the majority of RNA processing and bulk RNA turnover (Mackie, 2013). The C-terminal domain of RNase E interacts with RhlB (helicase), PNPase (polynucleotide polymerase and 3′ to 5′ exoribonuclease activities) and PAPI (poly(A) polymerase). Both PAPI and PNPase can add oligonucleotide tails (oligo(A) or A-rich, respectively) to the 3′ ends of RNAs following RNase E cleavage. This creates a single-stranded "landing pad" that promotes subsequent degradation by 3′-exonucleases (Khemici & Carpousis, 2004). In CLASH analyses, the 3′ ends of sequence reads will not generally correspond to *in vivo* cleavage sites because the RNA fragments are treated with RNase during library preparation. However, the presence of a non-encoded oligo(A) tract at the 3′ end of sequence reads is a clear indication that this represents a site that was cleaved and then oligoadenylated *in vivo*.

We previously reported that UV-crosslinking and high-throughput sequencing (CRAC) can be used to identify the binding sites for the RNA chaperone, Hfq, at base pair resolution in the model prokaryote *E. coli* and the related human pathogen, enterohaemorrhagic *E. coli* (EHEC) (Tree *et al*, 2014). These studies revealed that for many sRNA–mRNA interactions, the Hfq binding site is closely associated with the mRNA seed sequence. Formation of the sRNA–mRNA duplex at the Hfq binding site is predicted to induce dissociation from the single-stranded RNA binding site on the chaperone, providing directionality to the reaction (Tree *et al*, 2014). The endonuclease activity of RNase E is strongly stimulated by the presence of a free 5′ monophosphate on the substrate and a 5′ triphosphate therefore stabilizes newly synthesized mRNAs (Mackie, 1998). Recent work has demonstrated that sRNA–mRNA duplexes can guide RNase E cleavage of the mRNA by providing a free 5′ monophosphate to stimulate cleavage (Bandyra *et al*, 2012).

Together, these results indicated that formation of an sRNA–mRNA duplex may cause dissociation from Hfq and then direct RNase E cleavage of the mRNA. To test this model, we have identified targets of sRNA-mediated degradation transcriptome-wide and *in vivo* by applying CLASH to RNase E.

# Results

### UV-crosslinking identifies *in vivo* binding sites for RNase E

We reasoned that duplexed sRNA–mRNA pairs might be transiently associated with RNase E prior to mRNA degradation, allowing tagged RNase E to act as a bait in the capture of *in vivo* interactions by UV-crosslinking (CLASH) (Fig 1A). To facilitate affinity purification of RNA–RNase E complexes, the chromosomal copy of RNase E (*rne*) was C-terminally tagged with a tandem affinity His6-TEV cleavage site-FLAG tag (HTF). RNase E is essential for cell viability and was previously shown to retain function when C-terminally FLAG-tagged at the same site (Morita *et al*, 2005; Worrall *et al*, 2008). The strain expressing only RNase E-HTF was viable and showed normal processing of 9S rRNA precursor into mature 5S

---

**Figure 1. UV-crosslinking of RNase E reveals binding sites transcriptome-wide.**

A   Schematic of CLASH protocol for purification of RNA–RNA interactions. RNAs were UV-crosslinked to RNase E-HTF *in vivo* and purified using M2 anti-FLAG resin. RNAs were trimmed using RNase A/T1 and further purified under denaturing conditions. RNA linkers were ligated to the immobilized RNA–RNase E complexes. Duplexed RNAs may be ligated into a single contiguous molecule (left, CLASH) that gives information on RNA–RNA interaction occurring on RNase E. The remaining single RNAs reveal the site of RNase E binding within the transcriptome. Linker-ligated RNA–RNase E complexes were size-selected by SDS–PAGE and RNAs recovered for library preparation and sequencing. The schematic on the right represents the key steps in preparing UV-crosslinked RNA–protein complexes to map RNA–protein interactions sites (∼99% of reads recovered), and RNA–RNA interaction sites (∼1% of reads recovered). Colours correspond to key words in the flow diagram.

B   The 5′ UTR of *rne* is bound by RNase E and non-genomically encoded oligo(A) tails are maximally recovered −9nt from the *rne* start codon. Known stem loop structures (HP1–3) and the ribosomal binding site (RBS) are shaded grey.

C   RNase E binding and oligoadenylation of the *pldB-yigL* dicistronic transcript. The reported RNase E cleavage site (red dashed line) and SgrS binding site (grey shading) are indicated.

D   Length of non-genomically encoded oligo(A) tails recovered from RNase E-bound reads.

E   Position of Hfq binding and oligoadenylation relative to RNase E binding peaks. The cumulative position of Hfq and oligoadenylation peaks was determined relative to RNase E binding peaks for 672 RNase E binding sites that were within 1 kb of an Hfq binding peak. A detailed description of the data processing is presented in the Appendix Supplementary Methods.

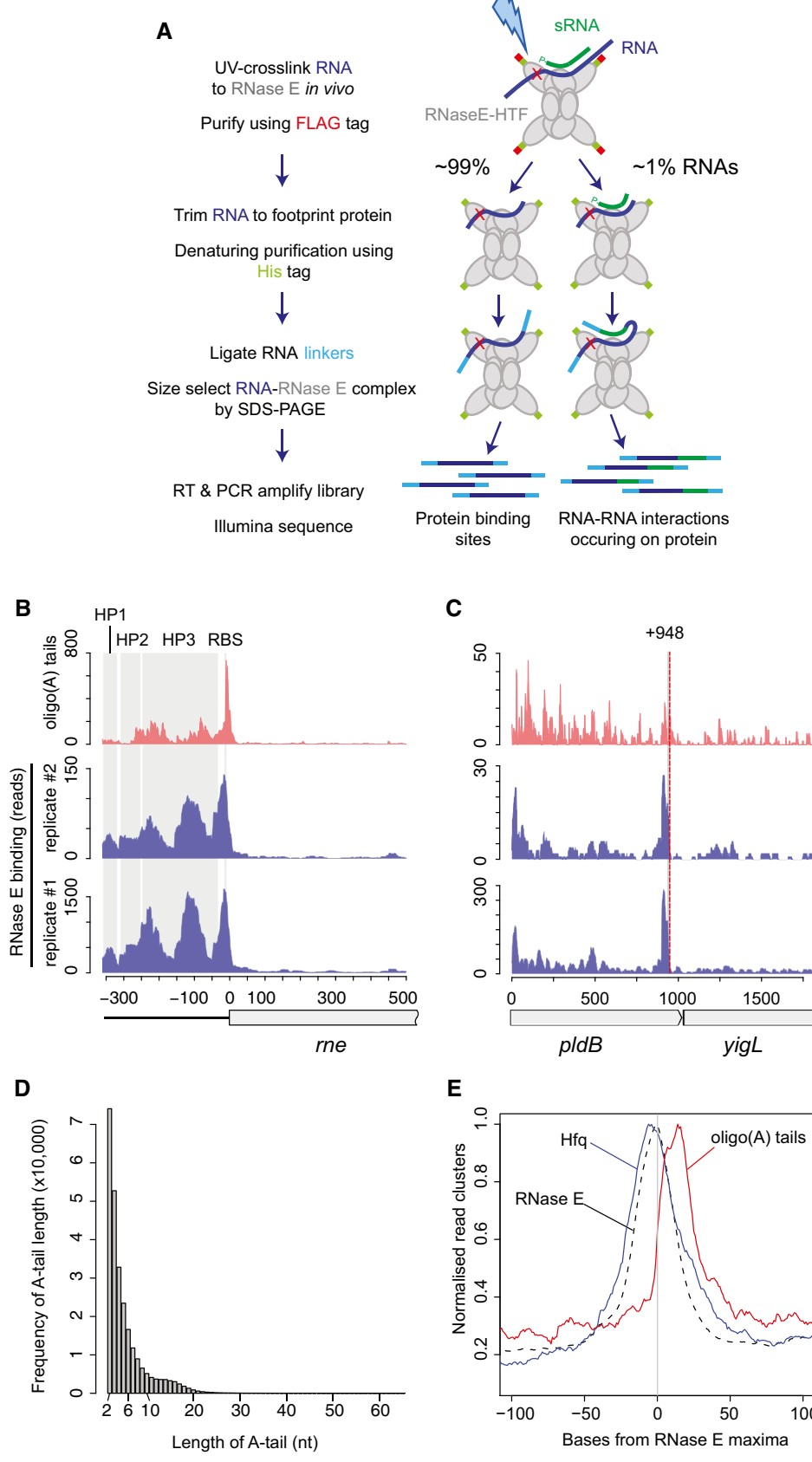

**Figure 1.**

rRNA (Ghora & Apirion, 1978), indicating that the fusion protein is functional (Fig EV1A). Following UV-crosslinking in actively growing cells, RNA–RNase E-HTF complexes were affinity-purified under denaturing conditions and crosslinked RNAs were trimmed using mild RNase A/T1 digestion. T4 RNA ligase was added to join RNase E-associated RNA duplexes into hybrid sequences, and to ligate Illumina sequencing compatible linkers to the ends of RNA fragments. Silver staining of eluates revealed co-precipitated proteins, with a clearly separated protein at the expected molecular weight of 118 kDa (Fig EV1B). We confirmed that this band was RNase E using LC-MS/MS. RNA–RNase E complexes were transferred to nitrocellulose, excised from the appropriate fragment of the membrane and recovered by protease digestion. Sequencing libraries were prepared by RT–PCR. Duplicate UV-crosslinking experiments showed a strong correlation in the number of reads mapping to individual transcripts (Spearman correlation = 0.97), and 79% of RNase E binding sites in dataset #2 (lower read depth) were also recovered in dataset #1. Sequence reads were mapped to the genome and represent sites of RNase E–RNA interaction (read statistics presented in Table EV1). Read clusters with > 10 reads were identified in 75% of annotated mRNAs, likely representing the repertoire of mRNAs expressed under our experimental conditions (Fig EV2), as RNase E is reported to be the primary factor responsible for initiating bulk mRNA turnover. In addition, close to 1% of reads were mapped to non-contiguous sites in the genome and represented RNA–RNA hybrid reads (see below).

As an initial step to verify our approach, we tested whether UV-crosslinking of RNA–RNase E complexes *in vivo* recovered known RNase E binding sites. Photocrosslinking experiments have demonstrated that RNase E autoregulates the stability of its own transcript (*rne*) by binding the hairpin structures HP1–HP3 within the 5′ UTR (Diwa *et al*, 2000; Schuck *et al*, 2009). We found that RNase E indeed binds to all three HP structures *in vivo*. Oligoadenylated reads, which are strongly indicative of endogenous 3′ ends (Khemici & Carpousis, 2004), peaked at −9 nts relative to the *rne* start codon, indicating that RNase E cleaves the *rne* transcript near the ribosomal binding site (Fig 1B). The small RNA SgrS binds *pldB* at +935−955 nt and stabilizes the *yigL* transcript by occluding an RNase E cleavage site at +948−955 nt within the dicistronic *pldB-yigL* mRNA (Papenfort *et al*, 2013). In agreement with this study, we find that RNase E binds 5′ of this cleavage site and overlaps the SgrS interaction site (Fig 1C). RNase E cleavage sites were recently mapped transcriptome-wide, identifying sites of 5′ monophosphate-independent ("direct entry") RNA cleavage (Clarke *et al*, 2014). We assessed RNase E binding at reported RNase E direct entry sites. Thirteen sites had > 50 reads within 200 nt of the direct entry cleavage site and ten showed a clear peak in RNase E binding or oligoadenylation at the direct entry site (Fig EV3). We conclude that our *in vivo* RNase E binding sites agree with published interactions and represent *bona fide* targets.

### Relationship between RNase E, Hfq and oligoadenylation sites

We previously reported that non-genomically encoded oligo(A) tails of 2–6 nt were present in 5% of Hfq-bound sequences (Tree *et al*, 2014). This indicates that Hfq binding sites are associated with endogenous 3′ ends that are oligoadenylated by PAPI. Oligo(A) tails were found in 0.7% of RNase E-bound reads and were predominately (76%) between 2 and 6 nt in length (Fig 1D). Hfq interacts with RNase E (Morita *et al*, 2005; Worrall *et al*, 2008), and sRNA interactions with an mRNA can facilitate RNase E recruitment and cleavage (Ikeda *et al*, 2011; Prévost *et al*, 2011; Bandyra *et al*, 2012). To gain insights into the arrangement of binding and cleavage sites, we compared the distribution of oligoadenylated sequences and Hfq crosslinking relative to RNase E binding sites. Maximal Hfq binding was cumulatively found five base pairs 5′ of the RNase E binding maximum (Fig 1E) although we note a significant overlap in these binding sites. In contrast, reads with oligo(A) tails, reflecting *in vivo* cleavage sites, were maximally recovered 13 base pairs 3′ of the peak in RNase E binding (Fig 1E).

These results support a model in which RNase E is frequently recruited to Hfq binding sites with a five base pair 3′-offset leading to RNA cleavage 13 nt downstream of the RNase E binding site and addition of a 2- to 6-nt oligo(A) tail. Recovery of more distant RNase E cleavage and oligoadenylation sites is limited by the length of the sequencing read. However, we note that our observations are consistent with *in vitro* characterization of the MicC–*ompD* interaction that directs RNase E cleavage 6 base pairs downstream of the sRNA–mRNA duplex (Bandyra *et al*, 2012).

### RNA–RNA interactions are recovered by RNase E-CLASH

In CLASH analyses, RNA duplexes that are bound by RNase E can be ligated together and recovered as cDNA sequencing reads that map non-contiguously to distinct sites in the transcriptome. These were identified and mapped using the Hyb software package (Travis *et al*, 2014). From 21.9 M mapped reads, we recovered 176,874 RNA–RNA interactions (0.8%, Tables EV1 and EV2) including 1,733 sRNA–mRNA interactions (Table EV3). There was substantial overlap between hybrids recovered in the two replicate datasets, and 41% of interactions identified in replicate #2 were also recovered in the larger replicate #1 dataset. We used the approach of Sharma *et al* (2016) to assess the theoretical false discovery rate expected from random ligation of RNAs in solution, and find that 58.8% of RNA–RNA interactions have an FDR < 0.05 (Table EV2 and Appendix Supplementary Methods).

To verify that RNase E-CLASH recovered *bona fide* sRNA–mRNA interactions, we looked for 125 experimentally verified sRNA–mRNA pairs within our datasets (Table EV4). Small RNA interactions were taken from sRNATarBase 3.0 (Wang *et al*, 2015), inspected for concordance with published sites and corrected where necessary (corrections to sRNATarBase 3.0 are presented in Table EV4). RNase E-CLASH analysis identified a statistically significant number of known sRNA–mRNA pairs (14/125, $P < 6.6 \times 10^{-4}$; Table EV5 and Appendix Supplementary Methods) including the sRNA–mRNA pair MicA–*ompA* (Fig 2A and B) (Rasmussen *et al*, 2005; Udekwu *et al*, 2005). We performed RNA-Seq on total RNA from EHEC and found that the recovery of hybrid reads was only weakly correlated with RNA abundance (Spearman correlation = 0.15; Fig EV4A), but was moderately correlated with RNase E crosslinking to single RNAs (Spearman correlation = 0.44; Fig EV4B). Similar results were found for the 125 known sRNA–mRNA interactions where hybrid recovery correlates more significantly with RNase E crosslinking (Spearman correlation = 0.15 for mRNA binding; Fig EV4C–F). Hybrid recovery is likely a function of both sRNA and mRNA association with RNase E, and we find a

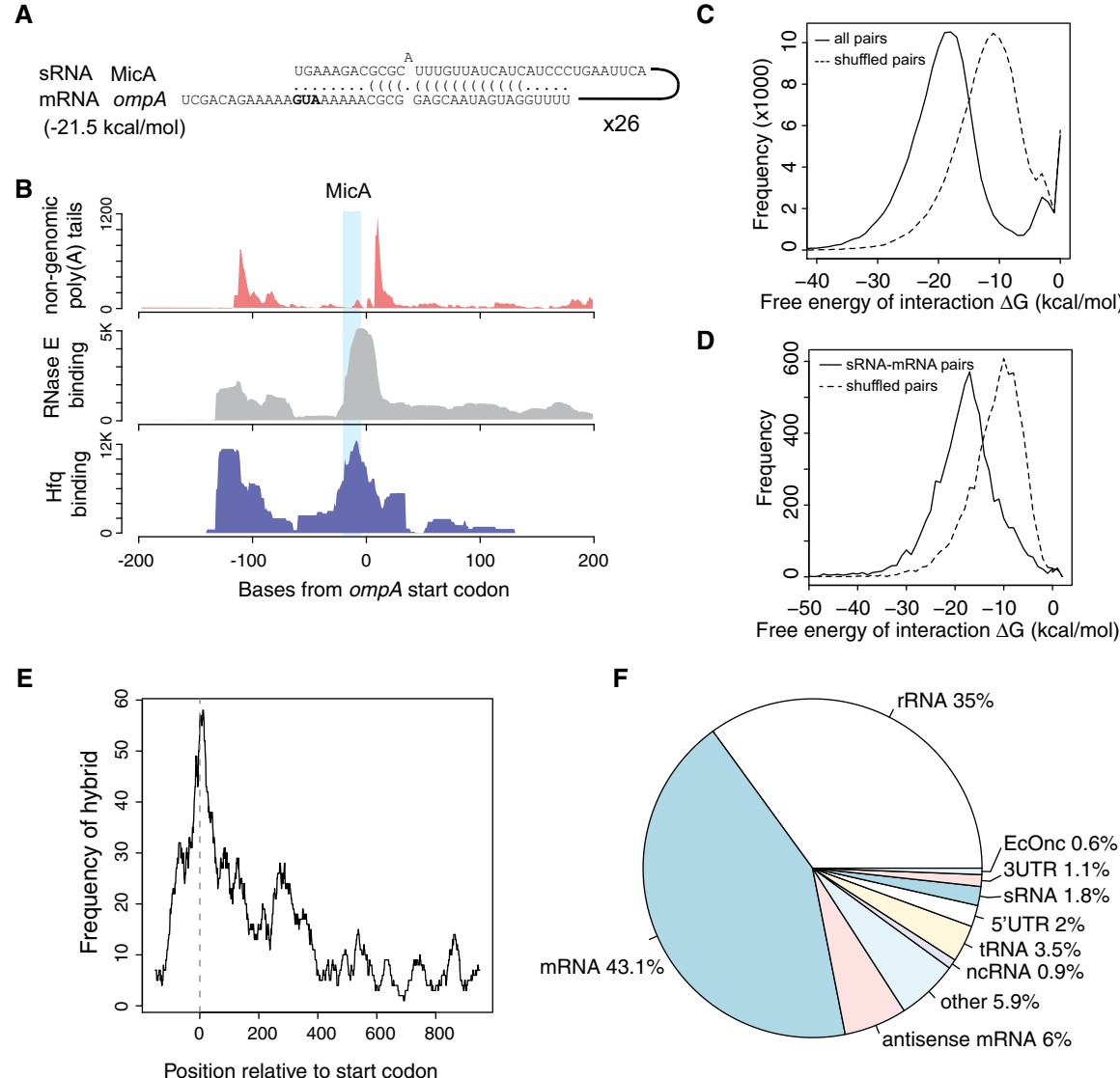

**Figure 2. Relative position, RNA class, and base pairing strength of small RNA hybrids recovered by RNase E-CLASH.**

A *In silico* folding of hybrid reads using the UNAfold suite of tools was used to predict base pairing and interaction strength between hybrid read halves. Reads mapping to the interaction between MicA and *ompA* mRNA are shown (ompA start codon in bold).

B Hfq binding (blue), RNase E binding (grey) and oligoadenylation (red) of the MicA binding site in the *ompA* mRNA. The MicA seed sequence is shaded blue.

C Interaction strength (Gibbs free energy) of all RNA–RNA interactions recovered by RNase E-CLASH (solid line) and for randomly paired hybrid read halves (dashed line).

D Interaction strength of sRNA–mRNA hybrid read halves (solid line) and randomly paired sRNA and mRNA read halves (dashed line).

E Position of sRNA–mRNA interactions relative to the mRNA start codon.

F Distribution of sRNA interactions recovered with each class of RNA.

general trend towards higher numbers of hybrid reads for known sRNA–mRNA interactions where both single RNAs were strongly crosslinked to RNase E (Fig EV4H). These results are consistent with hybrid reads being derived from RNA interactions on RNase E rather than from total cellular RNA. Small RNAs interact with mRNAs through base pairing, and hybrid reads generated from duplexed RNAs are predicted to have a lower-than-random free energy of interaction (ΔG) (i.e. greater stability). We compared the distribution of free energies for all RNA–RNA interactions identified (Fig 2C) and for sRNA–mRNA pairs (Fig 2D) with randomly paired hybrid read halves. The distribution of free energies from RNase

E-CLASH RNA–RNA interactions was significantly lower than for random pairs. These results are consistent with the hybrid sequences being derived from duplexed RNAs associated with RNase E.

Interactions between sRNAs and mRNAs that impair 30S ribosome binding and translation are generally positioned within a window extending from 50 nt upstream to 15 nt (five codons) downstream of the start codon (Bouvier *et al*, 2008). Binding sites for sRNAs identified by RNase E-CLASH were enriched within this window on mRNAs (Fig 2E), in agreement with 30S occlusion as a major pathway for sRNA function.

RNase E acts as a scaffold for the RNA degradosome and plays important roles in the degradation and processing of all RNA classes in *E. coli* (Mackie, 2013). We therefore determined the proportion of unique sRNA interactions that were contributed by each RNA class (Fig 2F). Messenger RNA coding regions and 5′ UTRs are characterized substrates for sRNA interactions and constituted 43.1 and 2% of interactions, respectively (reads that included sequences from both the 5′ UTR and CDS were categorized as CDS). The free sRNA pool can be "buffered" by sRNA–tRNA interactions (Lalaouna *et al*, 2015), which represented 3.5% of interactions in our dataset. In addition, sRNA interactions were recovered with rRNAs (35%) and other ncRNAs (6S, tmRNA, RnpB RNA, CsrB; 0.9%). Hybrids between different sRNA species were recovered, for both sRNAs encoded in the "core" genome (1.8%, 87 interactions) and pathogenicity islands (0.6%, 29 interactions), indicating an extensive sRNA–sRNA interaction network. These included the previously identified interaction between the bacteriophage-encoded anti-sRNA, AgvB, and the conserved core sRNA GcvB (82 unique hybrids) (Tree *et al*, 2014). Small RNAs can also be generated from the 3′ UTRs of mRNAs (Guo *et al*, 2014; Miyakoshi *et al*, 2015). 0.9% of hybrids with sRNAs mapped within 50 nt downstream of mRNA translation termination sites, potentially reflecting interactions involving 3′ UTRs or 3′ UTR-derived sRNAs. For all RNA classes presented in Fig 2F, the distribution of free energies of interacting RNAs was significantly lower than randomly paired hybrid halves ($P < 1 \times 10^{-9}$).

Our results indicate that sRNA–mRNA interactions recovered by RNase E-CLASH have significantly lower free energy than randomly paired RNA sequences and are predominately found close to the start codon, consistent with these hybrid sequences originating from *in vivo* sRNA–mRNA interactions. Numerous sRNA interactions were recovered with diverse ncRNA classes, including sRNA, rRNA, tRNA and other ncRNAs, revealing a complex network of sRNA interactions.

### Filtering functionally relevant RNA–RNA interactions

Proximity-dependent ligation protocols can potentially yield false-positive data through spurious ligation events, mapping artefacts or errors introduced during reverse transcription and PCR (Ramani *et al*, 2015). Since highly recovered interactions have a higher percentage of true positives (Ramani *et al*, 2015), ligation events can be weighted on the number of unique sequencing reads corresponding to individual interactions. We additionally used known and predicted attributes of sRNA–mRNA interactions to prioritize interactions for further analysis. This was based on (i) the number of unique sequence reads corresponding to the interaction; (ii) detection of the interaction in replicate datasets; (iii) recovery of the hybrid sequences in both RNA1–RNA2 and RNA2–RNA1 orientations, indicating ligation at opposite ends of the duplex; (iv) inclusion of a non-genomically encoded oligo(A) tail at the 3′ end of the target RNA sequence, which is indicative of sRNA-directed cleavage and subsequent tailing; and (v) overlap of both hybrid regions with Hfq binding sites determined by UV-crosslinking and indicating Hfq dependence (see Appendix Supplementary Methods). We confirmed that experimentally verified sRNA–mRNA interactions had a higher distribution of scores

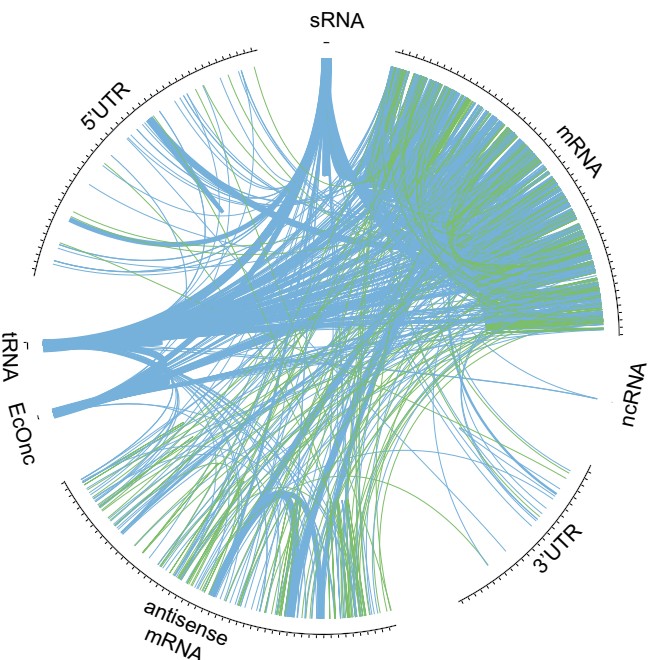

**Figure 3. RNase E-CLASH recovers RNA–RNA interactions between diverse RNA classes.**
RNA classes are labelled on the outer ring (rRNA has been omitted to clarify interactions between other RNA classes). RNA–RNA interactions with more than two unique hybrid sequences are presented. The thickness of the link represents the number of unique hybrid sequences recovered (up to a maximum of 50 sequences). RNA–RNA interactions where both hybrid halves overlap an Hfq binding site are coloured blue and non-overlapping interactions are coloured green.

compared to total sRNA–mRNA interactions recovered when applying these criteria (Fig EV5 and Appendix Supplementary Methods).

Strikingly, sRNA interactions that satisfied all five criteria, and were represented by multiple unique hybrid reads, were recovered for all RNA classes examined: mRNA, tRNA, rRNA, ncRNA, sRNA (both core and pathogen specific [EcOnc]) and mRNA antisense transcripts (Fig 3). The sRNA interactions with the most hybrid reads representing an interaction were with tRNA species and these interactions were also coincident with Hfq binding sites, indicating that tRNA is a major target for a subset of sRNAs.

Several characterized sRNAs target functionally related sets of mRNAs, allowing coordinated adaption of the transcriptome in response to specific challenges. Functionally related clusters of mRNA targets within an sRNA interactome may therefore constitute a further indication of reliability, as well as providing insights into the biological roles of the sRNAs involved. We therefore clustered functionally related sRNA interactions with a score of ≥ 1.1 using BiNGO (Maere *et al*, 2005) (Appendix Supplementary Methods). Consistent with previous reports (Sharma *et al*, 2011), targets for the core sRNA GcvB were enriched for mRNAs involved in branched-chain amino acid metabolism. The targets of seven other sRNAs showed significant enrichment of specific ontology classes (Table EV6). In particular, the EHEC-specific sRNA Esr41 (EcOnc14 in our earlier analysis) was significantly enriched for targets

annotated as "signal transduction". Esr41 bound three mRNAs with products involved in iron uptake: CirA (receptor for the iron-binding, catecholate siderophore), ChuA (haem receptor) and Bfr (bacterioferritin), which were analysed in more detail (see below). These results indicate that functionally related sRNA targets can be defined using gene ontology and are a further indicator of reliability.

## sRNA–RNA interactions define seed motifs

Within characterized sRNAs, a single "seed sequence" can initiate binding to multiple, distinct RNA targets. However, between sRNAs the seeds are heterogeneous in location and sequence, making them difficult to predict using only bioinformatic approaches (Peer & Margalit, 2011; Backofen *et al*, 2014). To identify putative, novel sRNA seed regions, we analysed sRNA–target RNA interactions. The base-paired nucleotides between each sRNA and target RNA were predicted by folding the hybrid read *in silico* using the UNAfold suite of tools. The base-paired nucleotides within the sRNA were plotted for each interaction (Fig 4 and Appendix Fig S1). Conserved sites of target base pairing were considered to be a seed region. Multiple seed regions were apparent in the sRNAs ChiX, RyhB, ArcZ, GadY, MgrR and Spot42. The motif discovery tool MEME (Bailey & Elkan, 1994) was then applied to identify conserved sequence motifs within target mRNAs that might be recognized by each sRNA seed. Highly enriched motifs were identified (e-value $< 10^{-4}$) within target RNAs for 12 sRNAs. GcvB was reported to recognize the consensus motif CACAaCAY in mRNAs through interactions with the GU-rich R1 seed region located at bases 66–89 (Sharma *et al*, 2011). We found that GcvB–target interactions were positioned within this R1 seed region (Appendix Fig S1D) and MEME identified the consensus motif ACAATAWC within GcvB-targeted RNAs that has complementary to bases 69–76 of the GcvB R1 seed region (Appendix Fig S1D). The consensus motif suggests that base G72 of GcvB frequently participates in G-U wobble interactions. For the 12 sRNAs with statistically significant target motifs, a complementary sequence was identified within the sRNA and likely represents a seed sequence (Fig 4A and B and Appendix Fig S1).

The seed sequence of the sRNA–mRNA pair MicC–*ompD* guides RNase E cleavage 6 nt downstream of the duplex (Bandyra *et al*, 2012). To determine whether this is a general phenomenon, we cumulatively analysed RNase E binding, oligoadenylation and Hfq binding relative to statistically significant seed motifs identified in target RNAs (Fig 4C–E). Oligo(A) tails were found to be maximally recovered 10 nt from the 3′ end of the seed motif (8-nt motif length) consistent with seed-directed RNase E cleavage. Hfq-bound reads were maximally recovered in the 10 nt 5′ to the seed motif, indicating that Hfq binding sites are often closely associated with the identified seed motifs.

Our results experimentally define seed motifs for sRNAs with multiple interactions and demonstrate that many sRNAs use more than one site for target RNA interactions. The newly identified sRNA seed motifs appear to direct RNase E cleavage and oligoadenylation of target RNAs at sites 3′ of the seed interaction.

## Functional testing of sRNA–mRNA interactions

To assess whether sRNA–mRNA interactions defined by RNase E-CLASH function in regulating gene expression, we used a two-plasmid system for monitoring translation of superfolder GFP fusions (Corcoran *et al*, 2012). Translational fusions were constructed for sRNA–mRNAs interactions with high scores, as defined above: *hdeA*-RyhB (score = 8.9), *zapB*-RyhB (7.6), *rssA*-RyeB (7.2), *frdA*-RyhB (6.7), *hdeA*-GadY (5.8) (Fig 5A–E), and for interactions with lower scores that were supported by the ontological analysis *chuA*-Esr41 (4.1), *cirA*-Esr41 (3.1) and *bfr*-Esr41 (4.2) (Fig 5F–H). Expression levels were reduced for all 8 of the fusions when co-expressed with the cognate sRNA. Mutations introduced into the mRNAs and sRNAs de-repressed the *frdA*-RyhB and *hdeA*-RyhB interactions, and all three Esr41 interactions. Point mutations in RyhB similarly relieved repression of *zapB*; however, synonymous mutations within the mRNA abolished expression and destabilized the transcript as assessed by qPCR (data not shown). A rare leucine codon was introduced into *zapB* by the M1 synonymous mutation, potentially explaining the poor translation of this mRNA. Introduction of compensatory mutations restored RyhB control of *frdA*, and Esr41 control of *chuA*, *cirA* and *bfr* verifying direct sRNA–mRNA interactions for these pairs and confirming that functional sRNA–mRNA interactions are recovered by the RNase E-CLASH method.

## The EHEC-specific sRNA Esr41 controls iron transport and storage

Our previous analysis of Hfq binding sites using UV-crosslinking identified numerous novel sRNAs within the pathogenicity islands of enterohaemorrhagic *E. coli*, referred to as EcOnc RNAs, but their RNA targets remained largely unknown (Sudo *et al*, 2014; Tree *et al*, 2014). The RNase E-CLASH dataset contained 810 unique hybrids with pathogenicity island-encoded EcOnc sRNAs identifying many target transcripts (Fig 3 and Table EV7). The EHEC-specific sRNA, Esr41 (EcOnc14 in our earlier analysis), was previously shown to affect the abundance of the *fliC* transcript and cell motility (Sudo *et al*, 2014). Here we have demonstrated that Esr41 regulates expression of the iron transport and storage proteins CirA, ChuA and Bfr (Fig 5F–H). The mRNA interactome of Esr41 is similar to the "core" genome-encoded sRNA, RyhB (Massé *et al*, 2005). We therefore additionally analysed translation of the *chuA*, *cirA* and *bfr* fusions in the presence of constitutively expressed RyhB (Fig 6A). Esr41 and RyhB repressed *bfr* to comparable levels, but Esr41 had a greater repressive effect on *chuA* translation, consistent with it base pairing closer to the *chuA* RBS. In contrast, Esr41 repressed *cirA* translation by 7.6-fold, whereas RyhB positively regulated *cirA* translation.

Esr41 is encoded on the pathogenicity island SpLE1 that also encodes the tellurite, phage and colicin resistance gene cluster *ter* (Whelan *et al*, 1997), and the enterobactin receptor Iha. Colicin 1A is a pore-forming toxin that uses the siderophore receptor CirA to enter the cell and cause bacterial cell death. RyhB confers sensitivity to colicin 1A through de-repression of CirA (Salvail *et al*, 2013), and we investigated the effect of Esr41 on colicin sensitivity. Constitutive expression of Esr41 conferred complete resistance to colicin 1A in the sensitive *E. coli* background, DH5α, but did not affect resistance in the EHEC background that is already colicin resistant (Fig 6 and data not shown). Deletion of *esr41* in EHEC strain ZAP198 conferred a fitness advantage in iron-limited medium (MEM-HEPES supplemented with 250 nM Fe(NO$_3$)$_3$ and

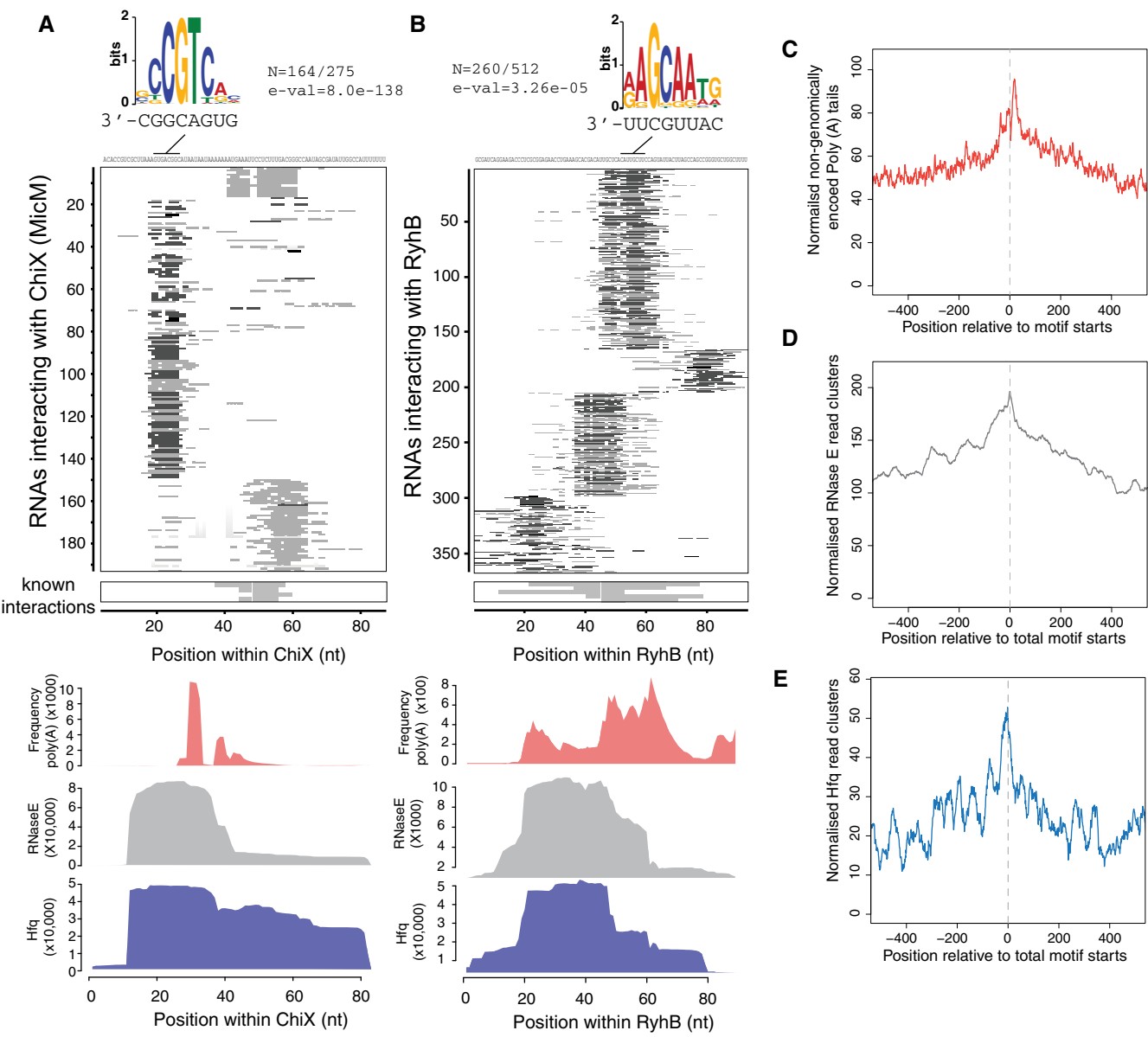

**Figure 4.  sRNA–RNA interactions identify sRNA seed sequences.**

A, B    (Top) Sequences interacting with ChiX (A) or RyhB (B) were analysed for conserved motifs using MEME. Motifs that were enriched in the target RNAs are shown above the heatmap with proportion of target RNAs carrying the motif (N) and the expected value for the motif (*e-val*). The complementary sequence motifs within ChiX or RyhB are shown below the logo. (Middle) Heatmaps showing the position of predicted base pairing for each interaction within ChiX or RyhB. Grey bars boxed below indicate the position of base pairing with experimentally verified mRNAs (known interactions). (Bottom) Hfq (blue) and RNase E (grey) binding sites within ChiX and RyhB. Hfq- and RNase E-bound sequence, and non-genomically encoded oligo(A) tails (red) within ChiX and RyhB are shown as line plots where the *x*-axis position correlates with heatmaps above.

C–E    Cumulative plots of oligoadenylation (C), Hfq (D) and RNase E (E) binding at predicted seed motifs.

0.1% glucose) consistent with repression of iron transporters by Esr41 (Fig 6E). Complementation of the *esr41* mutant by chromosomal knock-in of *esr41* restored the growth disadvantage to the *esr41* mutant.

These results demonstrate that, consistent with mRNA interactions identified by RNase E-CLASH, Esr41 regulates iron uptake and homeostasis in EHEC and can confer resistance to colicin 1A and colicin 1B in a sensitive background.

# Discussion

We demonstrate that interaction networks for bacterial sRNAs can be determined experimentally by UV-crosslinking sRNA–target RNA duplexes to RNase E. Our results revealed sRNA interactions with diverse RNAs including stable RNA species: rRNA and tRNA, other non-coding RNAs, and many different mRNAs. Here we have focused on the association of RNase E with sRNA–mRNA duplexes.

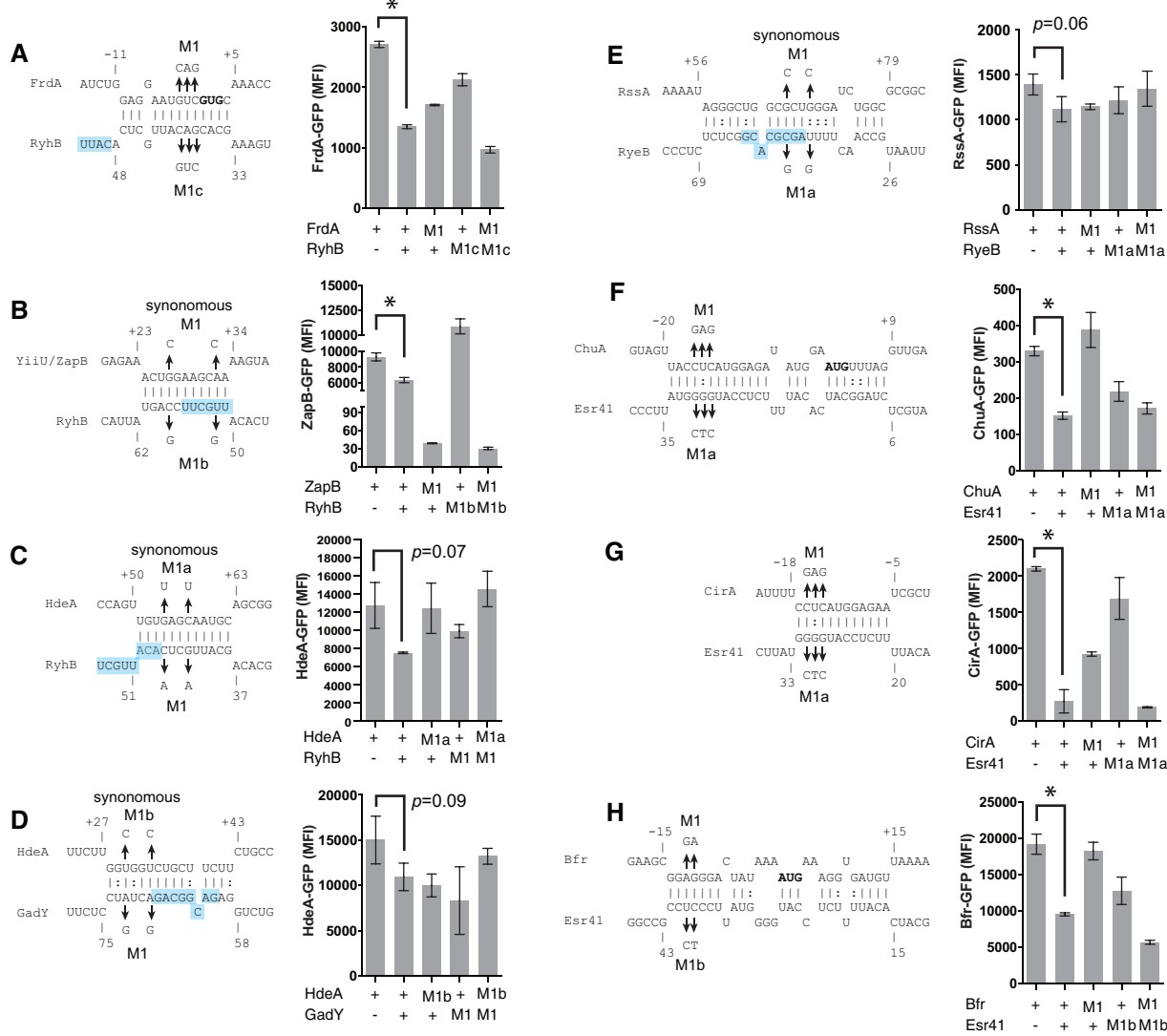

**Figure 5. Confirmation of high-scoring sRNA–mRNA interactions and interactions with the EHEC-specific sRNA, Esr41.**

A–H (Left) Base pairing patterns for the sRNA–mRNA pairs identified by CLASH were calculated using IntaRNA software (Busch *et al*, 2008) and are shown for *frdA*-RyhB (A), *zapB*-RyhB (B), *hdeA*-RyhB (C), *hdeA*-GadY (D), *rssA*-RyeB (E), *chuA*-Esr41 (F), *cirA*-Esr41 (G) and *bfr*-Esr41 (H). Point mutations and predicted sRNA seed sequences (blue shading) are indicated. (Right) Median fluorescence intensity (MFI) was assessed by FACS for mRNA-sfGFP fusions with compensatory base changes. In each case, introduction of a point mutation into an sRNA or mRNA construct (M1) is expected to reduce sRNA repression, which should be restored by combining complementary point mutants (last bar, M1-M1). A two-tailed *t*-test was used to calculate significance from biological triplicate cultures. Error bars represent SEM. *$P < 0.05$.

The CLASH analyses of RNase E-associated RNA duplexes recovered around 0.8% hybrids. This frequency is similar to that seen in previous analyses of human miRNAs associated with Argonaute 1 (Ago1) (Helwak *et al*, 2013) and double-stranded RNAs bound to Staufen (Sugimoto *et al*, 2015). In contrast, analysis of our previous Hfq UV-crosslinking data identified far fewer hybrids (~0.001% of mapped reads). Consistent with this finding, we previously found that many Hfq binding motifs overlap the mRNA seed sequence, suggesting that for these sRNA–mRNA interactions, duplex formation would likely dissociate the RNAs from Hfq (Tree *et al*, 2014). We therefore postulated that duplexes formed on Hfq are rapidly transferred to RNase E.

For a subset of sRNAs, we were able to define seed sequences within the sRNA and identify enriched motifs within target RNAs. Our analyses indicate that sRNAs commonly utilize multiple seed regions for target RNA base pairing. Target RNA seed sequences were closely associated with Hfq binding sites. This is consistent with our earlier model that duplex formation will render many Hfq binding motifs double stranded, promoting release of the base-paired RNAs and preventing re-binding to Hfq (Fig 4E). Oligoadenylation peaked 10 nt 3′ of the seed motif, indicating that many seed interactions direct cleavage of the mRNA and terminal nucleotide addition by poly(A) polymerase or PNPase (Fig 4C). This is consistent with *in vitro* results demonstrating RNase E cleavage of target

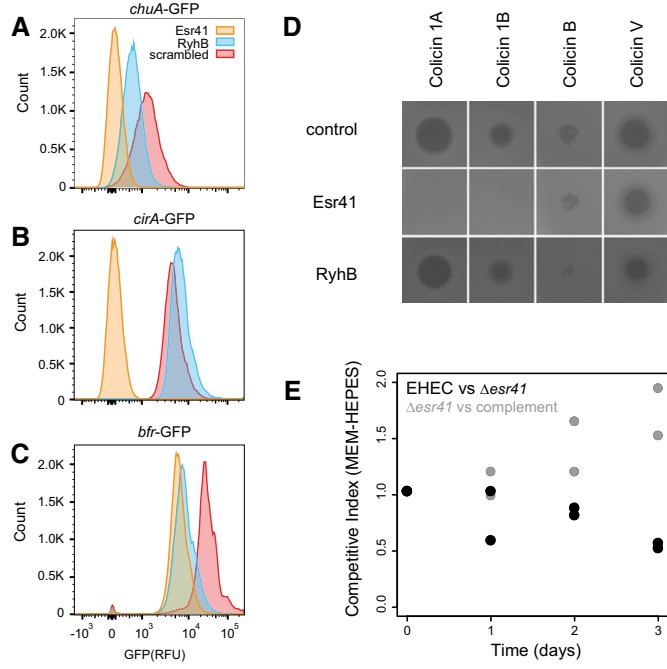

**Figure 6. The EHEC-specific small RNA Esr41 regulates iron uptake, storage and colicin resistance.**

A–C  FACS analysis of constitutively expressed sfGFP fusions to *chuA* (A), *cirA* (B) or *bfr* (C) in the presence of Esr41 (orange), RyhB (blue) or a scrambled RNA control (red). The histogram shows GFP fluorescence for each sRNA–mRNA fusion.

D  Esr41 confers resistance to colicins 1A and 1B in the sensitive background, *E. coli* DH5α. Top agar lawns of DH5α expressing a control scrambled RNA (pJV300), Esr41 (pZE12::*esr41*) or RyhB (pZE12::*ryhB*) were spotted with colicins indicated (top). Zones of clearing indicate sensitivity to the tested colicin.

E  Esr41 confers a competitive disadvantage on EHEC under iron-limiting conditions. Wild-type EHEC and isogenic Δ*esr41* strain (black), or EHEC Δ*esr41* and the chromosomally repaired strain EHEC Δ*esr41::esr41* were inoculated at equal densities and cultured in MEM-HEPES media for 3 days. The proportion of each strain was determined at each time point (days) and is expressed as a ratio relative to the starting inoculum where 1 is an equal fitness (see Appendix Supplementary Methods).

RNAs is guided to 5–6 nt 3′ of a duplexed 13-mer or sRNA (Bandyra *et al*, 2012).

The mechanism of sRNA-directed, RNase E cleavage has features in common with miRNA-directed cleavage by human Argonaute 2 (hAgo2). RNA targets that are fully complementary to the miRNA displace the PAZ domain of hAgo2 and induce a conformational change that results in cleavage of the miRNA–target duplex (Ameres *et al*, 2007; Wang *et al*, 2009). Thus, productive base pairing of the miRNA and target is sensed by competition between hAgo2 and the target RNA resulting in dissociation of the miRNA 3′ end. For the Hfq-RNase E complex, we suggest that sRNA–mRNA duplex formation at the Hfq binding motif dissociates the sRNA–mRNA pair from Hfq allowing interaction with RNase E and sRNA-directed cleavage of the target RNA 3′ of the seed motif.

A striking result from our RNase E-CLASH analysis was the range of RNA classes identified in RNA–RNA hybrids. The transcriptomes of both *E. coli* and *Salmonella* encode small RNAs embedded within mRNAs (Guo *et al*, 2014; Miyakoshi *et al*, 2015) lending weight to

the idea of a genomic palimpsest even in prokaryotes (Tuck & Tollervey, 2011) and potentially obscuring clear annotation of transcript classes. However, it is notable that all classes of RNA analysed were found in sRNA–RNA duplexes. We and others have identified small RNA species that act as sRNA sponges and this appears to be widespread. We recovered 152 unique sRNA–sRNA interactions in our CLASH data. These included our previously characterized interaction between the pathogenicity-associated sRNA AgvB and core sRNA GcvB (Tree *et al*, 2014). These results indicate that an extensive network of sponging interactions occur between sRNAs. Recent work demonstrated that sRNA interactions with tRNA spacer regions play important roles in "buffering" sRNA interactions to enhance specificity (Lalaouna *et al*, 2015). We identified 320 unique sRNA–tRNA interactions, including the previously reported RyhB–tRNA–Leu interaction (Lalaouna *et al*, 2015). We note that six sRNA–tRNA interactions contain > 10 nt of pre-tRNA sequence, indicating that minimally, these interactions occur before tRNA 5′ and 3′ maturation. Hfq has previously been shown to interact with tRNAs (Zhang *et al*, 2003; Lee & Feig, 2008; Tree *et al*, 2014), suggesting a role in facilitating sRNA–tRNA interactions. Extensive interactions of miRNAs with tRNA and rRNA have also been identified (Helwak *et al*, 2013) and it seems that these stable RNA species may act universally to buffer non-coding RNA interactions. These may stabilize sRNAs or miRNAs that are temporarily in excess over cognate targets and help prevent their inappropriate binding elsewhere.

The EHEC-specific sRNA Esr41/EcOnc14 was independently identified by Sudo *et al* (2014) and in our previous analysis of Hfq binding sites. We initially investigated the role of Esr41 in promoting colicin resistance through repression of CirA, and we were able to confirm that Esr41 confers complete colicin 1A and colicin 1B resistance when provided *in trans* in the colicin-sensitive background, DH5α. Colicin 1B is used by *Salmonella* Typhimurium to clear commensal *Escherichia coli* species (part of the normal flora) during gastrointestinal colonization (Nedialkova *et al*, 2014). Our results demonstrate that resistance to colicin 1B can be conferred by expression of a single, pathogen-specific small RNA. In contrast, the core genome-encoded sRNA RyhB promotes colicin 1A sensitivity through translational activation of CirA (Salvail *et al*, 2013).

Esr41 is encoded within a large pathogenicity island (SpLE1 or O-island 43/48) that confers colicin, tellurite and bacteriophage resistance, and also encodes the iron transporter/adhesin Iha. We were not able to test for decreased colicin 1A sensitivity in an EHEC Δ*esr41* strain due to the presence of the adjacent colicin resistance *ter* gene cluster. However, Esr41 targets identified by CLASH and confirmed by mutations included mRNAs encoding the iron transport and storage proteins ChuA, CirA and Bfr. A role in iron homeostasis is corroborated by competitive index experiments, demonstrating that deletion of *esr41* confers a fitness advantage to EHEC under relatively iron-limited conditions (250 nM Fe), indicating that Esr41 limits iron transport by repression of select iron receptors. The *Iha* gene is located upstream of Esr41 and encodes a receptor for the ferric iron-binding siderophore, enterobactin. We speculate that Esr41 is co-selected with Iha as Esr41-mediated repression of CirA (catecholate siderophore receptor), ChuA (haem receptor) and Bfr (bacterioferritin) would redirect iron transport through a pathway involving enterobactin and Iha, favouring maintenance of the O-island.

While this work was in revision, a related technique for sequencing sRNA–RNA interactions termed RIL-Seq was described (Melamed *et al*, 2016). This is conceptually similar to RNase E-CLASH, excepting that Hfq is used as a scaffold to capture sRNA–RNA duplexes and the purification is performed under native conditions as opposed to CLASH that uses a stringent purification protocol. Stringency is introduced into RIL-Seq analysis *in silico* where hybrid reads are filtered for statistical enrichment. We find a comparable number of statically significant sRNA–mRNA interactions are recovered by both techniques in log phase cells (633 using RIL-Seq and 782 using RNase E-CLASH) and similar sRNA seed regions and motifs are recovered for abundant sRNAs (e.g. ArcZ, MgrR, GcvB and CyaR), suggesting that both techniques capture *bona fide* sRNA–RNA interactions. Notably, the pools of RNA–RNA interactions recovered in association with Hfq and RNase E are expected to be different. RNase E processes a broad range of RNA species and is expected to associate with a subset of all sRNA–mRNA interactions that specifically result in target degradation.

We conclude that CLASH recovers functional RNA–RNA interactions when applied to RNase E in *E. coli,* allowing high-throughput identification of functional RNA targets for many sRNA species. A key advantage of this high-throughput approach is the ability to identify interactions that would not be predicted by extrapolating our current understanding of sRNA biology. We anticipate that profiling RNA interactions using CLASH will reveal diverse roles for both coding and non-coding RNAs in cell physiology.

# Materials and Methods

### Bacterial strains, plasmids and culture conditions

For CLASH analysis, *Escherichia coli* O157:H7 str. Sakai (GenBank Acc# NC_002695.1) was used to construct a dual-affinity-tagged HTF strain. Bacterial strains, plasmids and primers are presented in Table EV8. Strains were routinely grown on LB agar plates and broth supplemented with antibiotics where appropriate. For crosslinking and phenotypic experiments, *E. coli* O157:H7 was grown under virulence-inducing conditions in MEM-HEPES media (Sigma M7278) supplemented with 250 nM $Fe(NO_3)_3$ and 0.1% glucose.

### Preparation of CLASH sequencing libraries

Cells grown to OD 0.8 in MEM-HEPES (M7278) supplemented with 250 nM $Fe(NO_3)_3$ and 0.1% glucose were crosslinked with 1,800 mJ of UV-C. Cells were harvested by centrifugation at 4,000 *g* for 10 min, weighed and resuspended in 50 ml of ice-cold PBS. The cells were divided into 1 g pellets and snap-frozen in a dry ice/ethanol bath. One volume (1 ml/g) of lysis buffer [50 mM Tris–HCl (pH 7.8), 1.5 mM $MgCl_2$, 150 mM NaCl, 0.1% Nonidet P-40, 5 mM β-mercaptoethanol and 1 tablet "cOmplete" EDTA-free protease inhibitor (Roche)/50 ml] and 3 V of 0.1-mm zirconia beads were added to a cell pellet and vortexed 5 × 1 min with 1-min intervals on ice. Cell lysates were cleared by centrifugation (4,000 *g* for 20 min) and the supernatant was transferred to 1.5-ml microcentrifuge tubes and cleared at 16,000 *g* for a further 20 min. Supernatants were added to 200 μl of pre-washed M2 anti-FLAG resin

(Sigma-Aldrich) and incubated overnight. The resin was washed twice with 10 ml of TNM1000 (50 mM Tris–HCl pH 7.8, 1 M NaCl, 0.1% NP-40, 5 mM β-mercaptoethanol) and twice in 10 ml TMN150 (50 mM Tris–HCl pH 7.8, 150 mM NaCl, 0.1% NP-40, 5 mM β-mercaptoethanol), resuspended in 500 μl of TMN150 and incubated with 20–30 U of TEV protease for 2 h at 18°C. The slurry was centrifuged through a Bio-Rad Bio-spin column and the eluate collected. Approximately 500 μl of eluate was incubated with 0.15 U of RNace-IT (Agilent) at 20°C for 7 min. The digestion was stopped by the addition of 0.4 g of guanidine–HCl, 300 mM NaCl and 10 mM imidazole (pH 8.0). 100 μl of Ni-NTA slurry was pre-washed twice in 750 μl of wash buffer I (6 M guanidine–HCl, 50 mM Tris–HCl pH 7.8, 300 mM NaCl, 0.1% NP-40 and 5 mM β-mercaptoethanol). Eluates were added to the washed resin and incubated overnight at 4°C. The resin was washed twice with 750 μl of ice-cold wash buffer I and twice with 750 μl of 1× PNK buffer (50 mM Tris–HCl pH 7.8, 10 mM $MgCl_2$, 0.5% NP-40 and 5 mM β-mercaptoethanol). The eluates were transferred into a spin column (Pierce, Thermo Fisher, 69705). The subsequent reactions were performed in 80 μl reaction volumes on-column. 3′ ends were dephosphorylated by incubating for 45 min at 20°C with thermosensitive alkaline phosphatase (TSAP, Promega) and RNasin (Promega) in PNK reaction buffer (50 mM Tris–HCl pH 7.8, 10 mM $MgCl_2$ and 10 mM β-mercaptoethanol). The resin was washed once with 400 μl of wash buffer I and three times with 400 μl of 1× PNK buffer. The resin was incubated with tobacco acid pyrophosphatase (Epicentre) in 1× TAP buffer (Epicentre) and incubated at 20°C for 2 h, washed once with 400 μl of wash buffer I and then three times with 400 μl of 1× PNK buffer. The 5′ ends of bound RNAs were radiolabelled by phosphorylation with T4 PNK (4 μl, Sigma) and $^{32}$P-γATP (4 μl, PerkinElmer BLU502Z) in PNK reaction buffer for 100 min at 20°C, after which 100 nM of cold ATP was added and incubated for a further 50 min to complete 5′ end phosphorylation. The resin was washed once with 400 μl of wash buffer I and three times with 400 μl of 1× PNK buffer. To add 3′ linkers, the resin was incubated with 4 μl of T4 RNA ligase I (NEB) and 8 μl of miRCat-33 3′ linker (IDT) in PNK reaction buffer with 2 μl of RNasin (Promega) at 16°C for 16 h and then washed once with 400 μl of wash buffer I and three times with 1× PNK buffer. To add 5′ linkers, the resin was incubated with 4 μl of T4 RNA ligase I (NEB) and 1 μl of 100 μM 5′ linker (IDT; Table EV8) in PNK reaction buffer with 2 μl of RNasin (Promega) and 1 mM ATP at 16°C for 16 h. The resin was washed three times with wash buffer II (50 mM Tris–HCl pH 7.8, 50 mM NaCl, 10 mM imidazole, 0.1% NP-40, 5 mM β-mercaptoethanol). 200 μl of elution buffer (wash buffer II supplemented with 150 mM imidazole) was added to the resin and incubated at RT for 5 min. RNase E–RNA complexes were eluted into a clean microcentrifuge tube, and the elution was repeated. Complexes were precipitated with 100 μl of TCA and 40 μg of glycogen by incubating on ice for 30–60 min and centrifugation at 4°C for 20 min (16,000 *g*). Supernatants were removed and pellets washed with 800 μl of ice-cold acetone. Precipitate was centrifuged again at 16,000 *g*, supernatants were removed, and pellets were air-dried. The pellet was resuspended in 30 μl of 1× NuPAGE loading buffer. The sample was loaded onto a NuPAGE 4–12% Bis-Tris PAGE gel (Invitrogen) and run in MOPS SDS running buffer (Invitrogen). $^{32}$P-labelled RNase E complexes were transferred to a nitrocellulose membrane (Amersham Hybond ECL) by wet transfer using a Bio-Rad mini-Trans blot module in NuPAGE

transfer buffer (Invitrogen). Complexes were visualized by autoradiography using Kodak BioMax MS film and developed films realigned to the membrane. The high molecular weight complex (> 115 kDa) was excised from the membrane (see Fig EV1C). The labelled RNA was recovered by incubating the membrane fragment in 400 µl of wash buffer II supplemented with 1% SDS, 5 mM EDTA and 100 µg of proteinase K, for 2 h at 55°C. The supernatant containing labelled RNA fragments was transferred to a clean microcentrifuge tube. To precipitate the RNA fragments, 50 µl of 3 M NaOAc pH 5.2 and 500 µl of phenol:chloroform:isoamylalcohol was added, vortexed and centrifuged for 5 min at RT. The aqueous phase was transferred to a clean microcentrifuge tube and 1 ml of ice-cold EtOH and 20 µg of glycogen added. The precipitation was incubated at $-80°C$ for 30 min and centrifuged at 16,000 $g$ for 20 min, followed by a wash with 500 µl of ice-cold 70% EtOH and air-drying. The RNA pellet was resuspended in 13 µl of RT buffer I (miRCat RT oligo and 5 mM dNTPs) and reverse-transcribed using Superscript III as per the manufacturer's instructions. cDNA was amplified using Takara LA Taq, P5 and PE_miRCat PCR primers (Table EV8), and 2 µl of cDNA. cDNAs were amplified for 20–24 cycles to minimize bias in amplicons. 3–10 PCRs were pooled and ethanol-precipitated. PCR products were separated on a 3% metaphor agarose gel and smeared amplicons above primer dimers indicated in control samples were gel-extracted using a MinElute gel extraction Kit (Qiagen). Libraries were pooled and submitted for single-end 100-bp HiSeq2500 sequencing at GenePool (University of Edinburgh). Sequence data has been deposited at NCBI GEO (series GSE77463).

### Analysis of CLASH hybrids

Sequencing reads generated by RNase E-CLASH were analysed using the *hyb* package (Travis *et al*, 2014). Details of the *in silico* analysis are presented in Appendix Supplementary Methods.

### Confirmation of sRNA–mRNA interactions and phenotypic characterization of Esr41

We employed the two-plasmid system described by Corcoran *et al* (2012) to monitor translation efficiency of mRNA-sfGFP fusions. Plasmids containing small RNAs were cloned as described in Urban and Vogel (2007) excepting Esr41 was inserted into pZE12 using inverse PCR. Briefly, the mutagenic primers Esr41.ZE12.F and ZE12.5P.R were used to amplify a fragment of pZE12::*luc* that was DpnI-treated, gel-extracted and subsequently recircularized with T4 DNA ligase and transformed into DH5α. Clones containing an Esr41 insert were confirmed by sequencing. For mRNA fusions, clones were generated essentially as described in Corcoran *et al* (2012). Briefly, transcript start sites were identified using RegulonDB and the corresponding site in *E. coli* O157:H7 str. Sakai identified using BLAST. Primers were designed to amplify from the transcription start site to within the CDS encompassing the predicted region of sRNA–mRNA interaction (Table EV8). PCR products were cloned using NsiI and NheI (Fast digest enzymes, Thermo) and positive clones confirmed by sequencing. Point mutations were introduced using mutagenic primers listed in Table EV8 and confirmed by Sanger sequencing. Detailed methods for FACS and qPCR analysis of superfolder GFP fusions are presented in Appendix Supplementary Methods.

### Competitive index experiments

Indicated strains were grown overnight in LB at 37°C and the culture $OD_{600}$ adjusted to provide equal densities. Competing strains were inoculated at 1/1,000 into MEM-HEPES supplemented with 250 nM $Fe(NO_3)_3$ and 0.1% glucose. At 24-h intervals, the culture was diluted 1/1,000 in fresh media for a total of three subcultures (3-days growth). Cultures were diluted and plated onto LB plates to obtain well-separated colonies and 100 colonies were replica plated onto LB agar and LB agar supplemented with nalidixic acid (30 µg/ml) to select for marked strains. Competition experiments were repeated with nalidixic acid resistance and sensitivity in the opposite strain to account for any fitness cost associated with nalidixic acid resistance.

### Colicin sensitivity testing

Colicin 1A and B lysates were prepared from *E. coli* harbouring p3Z/Col1A and p3Z/ColB as described in Brickman and Armstrong (1996). Colicin 1B was prepared from *Salmonella* Typhimurium SL1344 by inducing with 1 µg/ml of mitomycin C and filtering the supernatant. Colicin V was prepared from *E. coli* strain NCTC50147 (Public Health England, UK) as described for colicin 1B. To test sensitivity to colicins, a top agar lawn of *E. coli* DH5α was prepared and 5 µl of colicin lysate spotted onto the lawn. Plates were incubated overnight at 37°C and scanned.

**Expanded View** for this article is available online.

## Acknowledgements

We thank Eric Masse for providing constructs for expressing colicins 1A and B. JJT and SAW were supported by funding from the Australian National Health and Medical Research Council (APP1067241). DLG, DT and SPM were supported by Wellcome Trust funding (WT090231MA) and research at the Roslin Institute is supported by BBSRC Institute grant funding (BB/J004227/1). DT was supported by Wellcome Trust funding (077248). Work in the Wellcome Trust Centre for Cell Biology is supported by Wellcome Trust core funding (092076). GK was supported by Wellcome Trust grant 097383 and by the MRC. MRW acknowledges funding from the Australian Government NCRIS scheme and the New South Wales State Government RAAP scheme.

## Author contributions

JJT, DLG and DT designed the experiments. SAW, JJT, SPM and KWL performed the experimental work. JJT, GK, NPD, IP and TGA analysed the data. All authors contributed to writing and editing the manuscript.

## Conflict of interest

The authors declare that they have no conflict of interest.

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
