## [Review Process File · The EMBO Journal]

Manuscript EMBO-2016-94639

Small RNA interactome of pathogenic *E. coli* revealed through crosslinking of RNase E

Shafagh Waters, Sean McAteer, Grzegorz Kudla, Ignatius Pang, Nandan Deshpande, Timothy Amos, Kai Wen Leong, Marc Wilkins, Richard Strugnell, David Gally, David Tollervey and Jai Tree

Corresponding author: Jai Tree, University of New South Wales

Review timeline:

Submission date:	26 April 2016
Editorial Decision:	24 May 2016
Revision received:	06 September 2016
Editorial Decision:	29 September 2016
Revision received:	06 October 2016
Accepted:	11 October 2016

Editor: Anne Nielsen

Transaction Report:

1st Editorial Decision

24 May 2016

Thank you for submitting your manuscript for consideration by the EMBO Journal. It has now been seen by three referees whose comments are shown below.

As you will see from the reports, the referees all express interest in the findings reported in your manuscript, although they also differ somewhat in their overall recommendations and ratings. In addition, the referees all point to a number of critical points related to data interpretation and analysis that will have to be fully addressed before they can support publication of a revised manuscript in The EMBO Journal. Given the discrepancy in the reviewer comments in conducted a round of cross-referee commenting and received the following input:

Ref #2:

Referee 1 notes that the manuscript is too short. I agree. The authors need to clarify a number of points and to explain better the biological significance of their results. Referee 3 has made a number of valid technical criticisms that need to be addressed. If the main point of the manuscript is that the authors have discovered new sRNAs and their targets in the EHEC strain, then I agree with referee 3 that the work is not of sufficient general interest for the audience of the EMBO J. I believe there are two major points to consider. 1) Does the work validate the claim that RNase E/CLASH is a generic method for discovering sRNA and their targets? 2) Does the work add anything new to our understanding of how sRNA and RNase E interact in targeting RNA for degradation?

Ref #1:

I agree fully with Referee #2's additional comment. The two major points described are spot-on. I think if the authors can provide additional support for the idea that this study provides a meaningful answer to EITHER of those two points, that the work will be important and broadly significant. With regard to Reviewer 3's statement that the methods are well established- while this is true, I think the real promise and impact of this study could be that we have a new tool for reliably and globally defining functionally-relevant RNA-RNA interactions in bacterial cells. This would address a rate-limiting step in research on bacterial RNA-based regulation. However, as stated in my review and others, I think the manuscript falls short of this promise as it stands.

Ref #3:

I fully agree with the synopsis by reviewer 1 - and as far as I can see her/his two questions are not answered at this point.

It is clear from these comments that there is great potential interest in your work both as a method for addressing RNA-RNA interactions in bacteria and a report of a number of new sRNA-target interactions. However, it is also clear that you will also have to extensively revise the current version to satisfy the referees. From our side, we would agree with referee #1 that either of the two key points raised by ref #2 will have to be addressed along with all technical/analysis issues. I realize that addressing all the referee concerns will require a lot of additional time and effort but given the overall interest from the referees we would be willing to give you the chance to do this in a revised manuscript. However, I would also understand if you would rather wish to publish the manuscript rapidly and without any significant changes elsewhere, in which case please let us know so we can withdraw it from our system.

For a revised manuscript for The EMBO Journal I would particularly ask you to focus your efforts on the following points:

- > Please extend the analysis to more conclusively call the number of known and new interactions captured. In addition, you will have to address the technical points from ref #3 about the overall enrichment and specificity of the RNA-RNA interactions mapped here
- > Please address the possible contribution from overall RNA abundance and provide a more extensive characterization of the binding sites observed
- > You will also see that the referees have a number of concerns about data description, control experiments and the resulting model (ref#2) that you will have to address.
- > In addition, the referees ask for more data/discussion on the nature of seed sequencing and its role in targeting substrate RNA

REFeree REPORTS

Referee #1:

This manuscript explores the use of UV-crosslinking, ligation and sequencing of hybrids (CLASH) as a high-throughput method for identification of RNA-RNA interactions in bacteria. The authors demonstrate that this method can successfully identify known small RNA-mRNA binding partners as well as a multitude of heretofore unknown RNA-RNA interactions. This method holds promise as a technology for broad-scale characterization of RNA-mediated regulation in bacteria. In this study, the authors provide evidence that the method identifies novel regulatory interactions, e.g., Esr41 with cirA and bfr. Follow up studies will no doubt explore the intriguing finding that the majority of small RNA interactions that are captured by CLASH are between small RNAs and other non-coding RNAs. This is a well-conceived study that reveals many new insights into the biology of RNA-mediated regulation in bacteria, even though it is primarily a methods paper. I have a few suggestions that I hope will help clarify certain aspects and strengthen the manuscript.

1. My primary criticism of the manuscript is that the depth of sequencing did not permit identification of very many known sRNA-mRNA regulatory interactions. Only 14/125 known sRNA-mRNA pairs were identified, and of these, 7 were represented by only 1 or 2 reads (Table S5). The authors use this

sparse data set to define criteria and assign priority scores that were then applied to validate other putative novel sRNA-mRNA interactions. I do not think this aspect of the analysis was very useful. In fact, of the putative new interactions chosen for further validation based on scores, only 1/5 (RyhB-frdA) was convincingly demonstrated as a bona fide interaction. They seemed to have higher accuracy using gene ontology as a method for prioritizing or selecting putative direct interactions, since 3/3 new targets for Esr41 were verified. This is the major weakness of the manuscript.

2. Was there a correlation between total abundance of particular sRNAs and mRNAs and their recovery as hybrids? For example, in total RNA, were MicA-ompA and MicM-chbC particularly abundant compared to RyhB-cysE or RyhB-shiA? Or not? Discussing this might shed light on whether there are other factors that influence efficiency of RNase E interaction with specific sRNA-mRNA pairs and provide new insight into molecular mechanisms of regulation.

3. Hybrid pairs for sRNAs and other RNA binding partners must be shown (e.g., in supplementary tables). 152 unique sRNA-sRNA interactions and 320 sRNA-tRNA interactions are discussed, but the data are not shown.

4. How did the sRNA interactions with other ncRNAs compare with the sRNA-mRNA pairs found? Free energy? Abundance?

5. p. 10, lines 1-18. The identification of seed sequences is not explained well in the text. It would help to provide some of the legend information in the text, it would make it much easier for the reader. Also, I assume each row of the heatmap represents the nucleotides of the sRNA that are interacting with the target represented in that row? How many targets are represented? Some sense of scale would be helpful.

Referee #2:

In this manuscript, the authors have performed a transcriptome-wide analysis of RNase E binding using UV crosslinking and high-density sequencing of cDNA. In addition, the technique (CLASH) detects rare events in which duplex RNA is ligated resulting in hybrids that permit the identification of RNA-RNA interactions. Novel sRNAs encoded in the pathogenicity islands of enterohaemorrhagic *E. coli* were identified. Predicted sRNA-mRNA interactions were validated with a functional test. In addition to detecting sRNA interactions with mRNA, a surprising number of other interactions were detected including sRNA-rRNA, sRNA-tRNA and sRNA-ncRNA duplexes. The results suggest that CLASH analysis of RNA bound to RNase E is a powerful tool for identifying sRNA and its targets.

1) The title of the article is imprecise. The interactome with the RNA degradosome includes more than RNase E interactions. PNPase and RhlB are also RNA binding proteins. 'Crosslinking of RNase E' or 'crosslinking of the RNase E component of the RNA degradosome' would be better.

2) Results, page 5, lines 18-21 and Supplementary Figure 1. The blot in the figure is not convincing since little precursor is detected relative to highly abundant 5S rRNA. A more sensitive test would be to probe the blot with an oligonucleotide specific to the 9S precursor of 5S rRNA.

3) Results, page 5, lines 26-29. As written, the sentence implies that the components of the RNA degradosome are not dissociated under denaturing conditions employing urea or guanidinium HCl. This is incorrect. His tagged RNase E can be denatured in 8 M urea and separated from other components of the degradosome as well as contaminating RNA by IMAC. In Worrall et al. 2008, mild non-denaturing treatment (0.5 M urea, 1 M salt) was used to strip RNA. In Morita et al. 2005, a scan of the article failed to find the words 'urea' or 'guanidinium'. All IP work in the Morita article was done under native conditions.

4) Results, page 5, lines 32-34. When the authors state that the 'sequence reads mapped predominantly to a single site', how did they handle mapping to the rRNA operons?

5) Results, page 6, lines 3-16. The RNase E binding result raises more questions than it answers. What is the specificity of these interactions? The entire 5' UTR of the *rne* message (361 nt) appears to interact with RNase E. Have the authors considered the possibility of multidentate interactions with tetrameric RNase E? The authors should cite Kenny McDowall's work mapping RNase E cleavage sites and comment on what is a correlation between binding and cleavage.

6) Results, page 9, lines 32-33. That 'gene ontology is a useful tool for identifying functionally related targets' seems like a trivial conclusion. The important question, which is addressed in the section on 'functional testing', is whether gene ontology is an indicator of reliability. This issue should be addressed in the section on 'functional testing'.

7) Results, page 10, lines 19-31. As an sRNA can have multiple seed sequences and these sequences are located at different positions relative to the 5' end, it is not clear why seed-directed RNase E cleavage would be optimally located 10 nt from the 3' end of the seed motif. Does this fit with the

model proposed in Bandyra et al. 2012? This issue should be addressed in the Discussion.

8) Discussion, page 12, lines 24-28. Together with previous work by the authors with Hfq (Tree et al. 2014), the model presented here seems to be at odds with the data. If duplex formation promotes dissociation of Hfq, RNase E binding, endonucleolytic cleavage, and then oligoadenylation, why are 5% of the Hfq sites in close proximity to non-coded oligo(A) sequences? Does this imply that Hfq remains bound to the duplex after RNase E binding?

9) Discussion, page 13, lines 20-28. The Lalaouna et al. 2015 reference demonstrates interaction between sRNA and tRNA spacers. When the authors mention '320 unique sRNA-tRNA interactions', are these with the spacer, the mature tRNA, or both? This point should be clarified.

10) Methods, page 14, lines 27-34. To my knowledge, the authors have never published the sequence of the HTF tag. Even if it is just a simple combination of the 6His tag, the optimal TEV sequence and the Flag tag, the authors should specify the exact sequence.

Referee #3:

In the manuscript "Small RNA interactome of pathogenic *E. coli* revealed through crosslinking of the degradasome" Waters et al. analyze RNA sequences recovered by UV crosslinking and pulldown of bacterial RNase E. They particularly focus on the small fraction of chimeric sequence reads which are interpreted as derived from bacterial small RNA-mRNA interactions and attempt to validate some of the predicted interactions using reporter assays. Finally they use *in vivo* assays that a pathogenic *E. coli* sRNA, Esr41 regulates colicin resistance of this bacterial strain. The approaches used for this study are well-established and thus, the main novelty resides in the characterization of the identified sRNA-mRNA interactions. However, my enthusiasm is dampened by what appears to be a somewhat superficial and not necessarily critical analysis of the high-throughput data in addition to the rather limited set of experiments addressing the biological implications of the predictions. While the results might be of potential interest to the field, the study appears to be a bit premature for publication in a journal of EMBO J's stature.

Specific comments/questions:

1. It is implicitly stated in the text that two replicates of the RNase E CLASH experiments were sequenced. Obviously the raw and the processed data should be deposited in a database, such as GEO. Perhaps a Supplementary table detailing how many reads per library were obtained, how many mapped, how many of the mappings were non-redundant, etc. Also, since part of the analysis also concerns RNase E binding sites (and not just the chimeras), a list of all binding sites would be helpful for the community as a resource. From the two replicates, what was the overlap of the RNase E binding sites (not just the chimeras)? Also, the authors claim that 41% of the hybrids found in the datasets were overlapping - it would be good to comment on what this means - is it the background that is overlapping?

2. The separation of signal from noise derived from copurification of abundant cellular RNAs is clearly challenging in all crosslinking/pulldown/sequencing approaches and different methods have been devised (e.g. for eCLIP, iCLIP, PAR-CLIP) to deal with the problem of abortive reverse transcription. In the original paper describing CRAC a characteristic deletion/insertion at the crosslinking site was described - it would be good to orient the reader whether this approach was employed for calling of bona fide binding sites for RNase E.

3. If I read it correctly, it appears that of 21.9 M mapped reads (in both replicates?), ~176k were chimeras and of those 1.7k were consistent with sRNA-mRNA interactions. It is difficult to see how the authors justify focusing on a fraction of reads representing 0.008% of the dataset without very stringent quality control (while virtually ignoring the rest). E.g. in comparison to the Hfq CRAC experiment - how many of those reads were recovered? Most importantly, what do the other 175k chimeric reads represent, why are they in such an excess, given that - if I understand it correctly - RNase E does not necessarily interact with dsRNA all by itself but is guided to sRNA-mRNA pairs? How many of these chimeras would one expect by chance (perhaps some analysis in the style of that presented in the Grosswendt et al paper cited would be helpful)? Also, the authors demonstrate that they recover 14 of 125 predicted sRNA-mRNA pairs (what was the basis for limiting the predicted set to 125 - the database contains >700 interactions) - nevertheless, they recover 1700 sRNA-mRNA interactions by themselves, which means that either only ~1% of the possible sRNA-mRNA

interactions were previously predicted, or that the CLASH approach calls more interactions than are practicable.

4. I did not notice whether the authors compared their data to RNAseq datasets - what is the number of mRNAs expressed, what is the number of sRNAs expressed, how abundant are they, how many of those are recovered by CLASH, etc.

5. Apparently, most of the sRNA interactions involved other ncRNA molecules. It would be good to comment on the significance of this finding. What is the basis for the interpretation that this is a bona fide interaction and not the sequencing of fragments of the most abundant cellular RNAs? It would be anticipated that these sites would also be found in the Hfq CLASH/CRAC experiments given that Hfq apparently hands over the sRNA/RNA target duplex to RNase E. Does binding of RNase E to these molecules (tRNA/rRNA) result in RNase cleavage?

6. Figure 3 is left almost uninterpreted and to the imagination of the reader.

7. Figure 4: in panel B - the seed sequence is identified in ~20% of the putative targets - what other sequence are these targets interacting with. Also, the authors claim that the RyhB sequence GCTCACAT is a seed - even with two mismatches to the CTGGAAGC identified - it would be good to include a discussion about the stringency of the seeds - if in the end 4-5 nt are enough for interaction, it becomes very difficult to predict binding sites.

8. Figure 5: Of the set of 8 high-scoring sRNA-mRNA pairs, it appears that only 4 interactions can be validated (panels A,F,G,H) - the other four either show very little sRNA dependence (B,D,E), or incomplete rescue using compensatory mutations. What is the confidence in the assay? Perhaps it would be good to show some confidence measure for the changes.

9. Figure 6: Here the genetic interaction of Esr41 (and RyhB) with the *chuA*, *cirA*, and *bfr* genes is analyzed in vivo, and presence of Esr41 represses the genes. Nevertheless, because the authors don't show that this repression is dependent on the direct sequence-specific interaction with Esr41 (e.g. by mutating the binding sites, which the authors have in their hands after all), any of these Esr41 dependent effects could still conceivably be indirect.

1st Revision - authors' response

06 September 2016

We thank the referees for the careful consideration that has gone into the reviews. We have performed multiple additional analyses in response to the helpful suggestions provided. We hope that these will serve to confirm that the approach described demonstrates that CLASH with RNase E is indeed a generic method for discovering sRNA and their targets. The principal insights provided into the interaction of sRNAs with RNase E arise from the range and diversity of sRNA targets that are bound by RNase E. We hope that these key outcomes are now sufficiently established for acceptance of the revised MS.

Referee #1:

This manuscript explores the use of UV-crosslinking, ligation and sequencing of hybrids (CLASH) as a high-throughput method for identification of RNA-RNA interactions in bacteria. The authors demonstrate that this method can successfully identify known small RNA-mRNA binding partners as well as a multitude of heretofore unknown RNA-RNA interactions. This method holds promise as a technology for broad-scale characterization of RNA-mediated regulation in bacteria. In this study, the authors provide evidence that the method identifies novel regulatory interactions, e.g., Esr41 with *cirA* and *bfr*. Follow up studies will no doubt explore the intriguing finding that the majority of small RNA interactions that are captured by CLASH are between small RNAs and other non-coding RNAs. This is a well-conceived study that reveals many new insights into the biology of RNA-mediated regulation in bacteria, even though it is primarily a methods paper. I have a few suggestions that I hope will help clarify certain aspects and strengthen the manuscript.

1. My primary criticism of the manuscript is that the depth of sequencing did not permit identification of very many known sRNA-mRNA regulatory interactions. Only 14/125 known sRNA-mRNA pairs were identified, and of these, 7 were represented by only 1 or 2 reads (Table S5). The authors use this sparse data set to define criteria and assign priority scores that were then applied to validate other putative novel sRNA-mRNA interactions. I do not think this aspect of the analysis was very useful. In fact, of the putative new interactions chosen for further validation based on scores, only 1/5 (RyhB-*frdA*) was convincingly demonstrated as a bona fide interaction. They seemed to have higher accuracy using gene ontology as a method for prioritizing or selecting putative direct interactions, since 3/3 new targets for *Esr41* were verified. This is the major weakness of the manuscript.

The reviewer is correct in their conclusion that many sRNA-mRNA interactions are represented by a single hybrid read. Similar results have been reported for CLASH analysis of miRNA (Helwak Cell 2014) where 48.1% of miRNA-mRNA interactions are represented by a single interaction. Grosswendt et al (Mol Cell 2014) also found that 81.3% of miRNA:targets are supported by a single hybrid. Importantly, the later demonstrate that single hybrid interactions display essentially the same features (seed matches, conservation, and T>C conversions) as those with >1 hybrid. We find the 77.7% of sRNA-mRNA interactions are supported by a single read. Notably, we confirmed that Esr41 interactions with Chua and CirA, which were each supported by a single hybrid, are functional.

Recovery of a small percentage of known interactions is also consistent with both studies on miRNA:target capture. Grosswendt recovered 149 interactions reported from 369,030 (0.0004%) interactions on miRTarBase (human, mouse, C. elegans, EBV, KSV). Helwak et al recover 77 known miRNA interactions (32 represented by a single hybrid). Our recovery of 11.2% of known interactions compares favorably with these studies. For our data we have calculated the probability of recovering a known mRNA seed sequence with an sRNA at $p \ll 6.6 \times 10^{-4}$ (calculation presented in Supplementary Methods: Hybrid filtering). Collectively, we feel that our recovery of sRNA-mRNA hybrids is consistent with previous work on miRNA interactions and that we find a highly statistically significant number of known interactions that support our dataset given the relatively limited database of experimentally verified sRNA-mRNA interactions. We would also note that the expression profile of mRNAs and sRNAs in EHEC grown in virulence inducing MEM-HEPES is likely quite different to the majority of studies carried out in E. coli K12.

We have defined criteria for scoring sRNA-mRNA interactions based on the current understanding of sRNA interactions (Hfq binding, mRNA cleavage and polyadenylation) and previously reported indicators of reliability in CLASH/hiCLIP/RPL-Seq/PARIS/SPLASH/LIGR-Seq data (number of hybrids, recovery of cDNAs in both orientations). While we agree that the number of known interactions is limited, we have very few indicators of true positives available to assess our ranking system and our current analysis meets statistical significance. We feel that ranking the interactions is essential to allow the reader some idea of whether an interaction has the expected hallmarks of a bona fide sRNA-mRNA interaction.

2. Was there a correlation between total abundance of particular sRNAs and mRNAs and their recovery as hybrids? For example, in total RNA, were *MicA-ompA* and *MicM-chbC* particularly abundant compared to *RyhB-cysE* or *RyhB-shiA*? Or not? Discussing this might shed light on whether there are other factors that influence efficiency of RNase E interaction with specific sRNA-mRNA pairs and provide new insight into molecular mechanisms of regulation.

*We have now performed total RNA-Seq on our EHEC strain grown in MEM-HEPES and looked at correlates with hybrid recovery for the 125 known interactions and the total hybrids datasets (new Supplementary Figure 5). We find that hybrid recovery (all interactions) correlates weakly with RNA abundance (Supplementary Figure 5A, Spearman's correlation = 0.14) and moderately with RNase E crosslinking (Supplementary Figure 5B, Spearman's = 0.44). These results are consistent with hybrids being derived from RNaseE bound RNA-interactions rather than total cellular RNA. For known interactions we find that hybrid recovery correlates weakly with RNaseE crosslinking to the mRNA (Supplementary Figure 5D, Spearman's = 0.15). We have also looked at correlations with sRNA classes defined by Schu et al EMBO (2015) and note that more hybrids are recovered for higher R16/K31 ratios (class I < 1 < class II), but there is no statistical correlation (Supplementary Figure 5G). The number of hybrids recovered is likely a function of RNaseE crosslinking to both interacting halves and we see a general trend towards higher hybrid numbers when both RNAs crosslink strongly to RNase E (interactions from Supplementary Table 6 plotted in Supplementary Figure 5H). In the case of *ChiX(MicM)-chbC*, the abundant crosslinking of *ChiX* appears to compensate for the low abundance of *chbC*.*

We have added these results to pg 7 lines 36-37 and pg 8 lines 1-11.

3. Hybrid pairs for sRNAs and other RNA binding partners must be shown (e.g., in supplementary tables). 152 unique sRNA-sRNA interactions and 320 sRNA-tRNA interactions are discussed, but the data are not shown.

All RNA-RNA interactions are provided in Supplementary Table 2 with detailed annotations of the transcript names, RNA classes, sequence, Hfq binding data, interaction strength of RNAs (ΔG), and number of hybrids recovered. We have duplicated information in this table for sRNA-mRNA interactions (Supplementary Table 3) and EcOnc-mRNA interactions (Supplementary Table 8) for ease of reference, but feel that there are too many subclasses of RNA-RNA interactions to duplicate them all in separate tables.

4. How did the sRNA interactions with other ncRNAs compare with the sRNA-mRNA pairs found? Free energy? Abundance?

The distribution of sRNA interactions recovered with different RNA classes is presented in Figure 2F. The distribution of free energy for RNA classes interacting with sRNAs is less than random interaction strength (from shuffled interaction pairs) (Figure below, $p < 1e-8$ for all RNA classes). We have added the lines "For all RNA classes presented in Figure 2F the distribution of free energies of interacting RNAs was significantly lower than randomly paired hybrid halves ($p < 1 \times 10^{-9}$)." (pg 9 lines 7-9).

We are currently investigating the functional significance of some of these sRNA-RNA interactions and feel that a separate publication is required to fully investigate them. We do recover 36 unique hybrids that map to the recently reported sponging interaction between RyhB and the ETS of tRNA-Leu (Lalalouna et al Mol Cell 2015), suggesting that at least a subset of these non-canonical interactions will be functional.

5. p. 10, lines 1-18. The identification of seed sequences is not explained well in the text. It would help to provide some of the legend information in the text, it would make it much easier for the reader. Also, I assume each row of the heatmap represents the nucleotides of the sRNA that are interacting with the target represented in that row? How many targets are represented? Some sense of scale would be helpful.

We have added x- and y-axes to the heatmaps to number the interacting RNAs (y-axis) and the position within ChiX and RyhB (x-axis). We have edited the text in this section to more clearly present our results and fold in some of the information in the Figure legend.

Referee #2:

In this manuscript, the authors have performed a transcriptome-wide analysis of RNase E binding using UV crosslinking and high-density sequencing of cDNA. In addition, the technique (CLASH) detects rare events in which duplex RNA is ligated resulting in hybrids that permit the identification of RNA-RNA interactions. Novel sRNAs encoded in the pathogenicity islands of enterohaemorrhagic *E. coli* were identified. Predicted sRNA-mRNA interactions were validated with a functional test. In addition to detecting sRNA interactions with mRNA, a surprising number of other interactions were detected including sRNA-rRNA, sRNA-tRNA and sRNA-ncRNA duplexes. The results suggest that CLASH analysis of RNA bound to RNase E is a powerful tool for identifying sRNA and its targets.

1) The title of the article is imprecise. The interactome with the RNA degradosome includes more than RNase E interactions. PNPase and RhlB are also RNA binding proteins. 'Crosslinking of RNase E' or 'crosslinking of the RNase E component of the RNA degradosome' would be better.

The title has been amended to "Small RNA interactome of pathogenic E. coli revealed through crosslinking of RNase E".

2) Results, page 5, lines 18-21 and Supplementary Figure 1. The blot in the figure is not convincing since little precursor is detected relative to highly abundant 5S rRNA. A more sensitive test would be to probe the blot with an oligonucleotide specific to the 9S precursor of 5S rRNA.

We have now blotted for 9S in our tagged and untagged strains and the results are much clearer. Our HTF tagged strain does not have impaired 5S rRNA processing. The new blot replaces the 5S blot in Supplementary Figure 1.

3) Results, page 5, lines 26-29. As written, the sentence implies that the components of the RNA degradosome are not dissociated under denaturing conditions employing urea or guanidinium HCl. This is incorrect. His tagged RNase E can be denatured in 8 M urea and separated from other components of the degradosome as well as contaminating RNA by IMAC. In Worrall et al. 2008, mild non-denaturing treatment (0.5 M urea, 1 M salt) was used to strip RNA. In Morita et al. 2005, a scan of the article failed to find the words 'urea' or 'guanidinium'. All IP work in the Morita article was done under native conditions.

The reviewer is correct, Morita et al was mistakenly copied from the references on line 19. We have now used LC MS/MS to confirm that the protein above 98 kDa is RNase E and include this confirmation in the Results section (pg 5 lines 29-32). We were not able to conclusively identify the co-precipitated proteins although we find some components of the degradosome in excised gel fragments. It is currently unclear why these proteins are retained during our 6M guanidinium purification, but may be dependent on the crosslinked RNA. As we excise the RNA-RNase E containing section of the SDS-PAGE gel after transfer to a membrane, these co-purified proteins do not affect our conclusions regarding RNase E targeting and recovery of RNA-RNA interactions.

4) Results, page 5, lines 32-34. When the authors state that the 'sequence reads mapped predominantly to a single site', how did they handle mapping to the rRNA operons?

We agree that this is ambiguous and have amended the text to "Sequence reads were mapped to the genome and represent sites of RNase E-RNA interaction". Our intention was to distinguish mapping of non-hybrid reads (>99% of mapped reads) from hybrid reads. Those non-hybrid reads that mapped to transcripts encoded in multi-copy (eg: rRNA, tRNA) were randomly assigned to a single copy of the transcript using the -r Random flag of novoalign.

5) Results, page 6, lines 3-16. The RNase E binding result raises more questions than it answers. What is the specificity of these interactions? The entire 5' UTR of the rne message (361 nt) appears to interact with RNase E. Have the authors considered the possibility of multidentate interactions with tetrameric RNase E? The authors should cite Kenny McDowall's work mapping RNase E cleavage sites and comment on what is a correlation between binding and cleavage.

We have not been able to define a sequence or structural motif that recruits RNase E to an RNA substrate. Many of the binding “peaks” (like rne and pldB-yigL shown) extend across the RNA and it seems plausible that the RNase E tetramer (~472 kDa) could make extensive contacts with the RNA or acts in a processive manner. We cannot distinguish multidentate contacts from multiple, independent binding events that may occur in successive rounds of processing.

We have now looked at the sites of RNaseE direct entry mapped by Kenny McDowall’s group. Our experimental conditions and model organism are different, but we have been able to identify 13 sites that give a reasonable read depth (>50 reads) within 200nt and have looked at RNaseE binding and polyadenylation at these sites. For all direct entry sites except yncL, nuoA, and metL, we find a prominent peak in RNaseE binding or (for yf1K) oligoadenylation at the direct entry site (new Supplementary Figure 4). We have included a reference to Clarke et al 2014 and comment on the correlation between our UV-crosslinking data and direct entry sites identified by these authors (pg 6 lines 19-24).

6) Results, page 9, lines 32-33. That 'gene ontology is a useful tool for identifying functionally related targets' seems like a trivial conclusion. The important question, which is addressed in the section on 'functional testing', is whether gene ontology is an indicator of reliability. This issue should be addressed in the section on 'functional testing'.

We have modified the text to refer the reader to functional testing of the Esr41-target interactions and added the sentence: “These results indicate that functionally related sRNA targets can be defined using gene ontology and are a further indicator of reliability.”

7) Results, page 10, lines 19-31. As an sRNA can have multiple seed sequences and these sequences are located at different positions relative to the 5' end, it is not clear why seed-directed RNase E cleavage would be optimally located 10 nt from the 3' end of the seed motif. Does this fit with the model proposed in Bandyra et al. 2012? This issue should be addressed in the Discussion.

This fits well with the model proposed by Bandyra et al 2012. These authors find that a 5' monophosphorylated 13-mer or MicC sRNA can direct RNase E cleavage 5-6 nt downstream of the duplexed nucleotides. Given that we are measuring from predicted seed motifs rather than the actual duplexed nucleotides (that may extend further than the seed), we feel that our observed 10nt spacing from 3' of the motif to the oligoadenylation peak is in excellent concordance with these in vitro results. We have added the line “Oligoadenylation peaked 10 nt 3' of the seed motif and indicates that many seed sequences direct cleavage of the mRNA (Figure 4C) consistent with in vitro results demonstrating RNase E cleavage of target RNAs is guided to 5-6 nt 3' of a duplexed 13-mer or sRNA (Bandyra et al. 2012)” to the discussion (line 26-30, pg 13).

8) Discussion, page 12, lines 24-28. Together with previous work by the authors with Hfq (Tree et al. 2014), the model presented here seems to be at odds with the data. If duplex formation promotes dissociation of Hfq, RNase E binding, endonucleolytic cleavage, and then oligoadenylation, why are 5% of the Hfq sites in close proximity to non-coded oligo(A) sequences? Does this imply that Hfq remains bound to the duplex after RNase E binding?

Our model of sRNA-mRNA duplex dissociation is supported by i) overlap of Hfq binding sites and duplexed seed sequences, ii) overlap of RNase E and Hfq binding sites, and iii) previous in vitro work demonstrating duplex dissociation (Fender et al 2010; Hopkins et al 2011; Lease and Woodson 2004; Updegrave et al 2008). Using oligo(A) tails as an indicator of sequential binding of Hfq > RNase E > PAPI is confounded by observations from the Hanjnsdorf lab that show Hfq preferentially binds oligo(A) tailed RNAs and stimulates PAPI extension of these tails (Hanjnsdorf & Regnier PNAS 2000; Folichon et al FEBS J 2005). It seems plausible that cleaved/oligoadenylated RNAs re-associate with Hfq during polyadenylation and degradation (Hfq > RNase E > PAPI > Hfq > PAPI).

9) Discussion, page 13, lines 20-28. The Lalaouna et al. 2015 reference demonstrates interaction between sRNA and tRNA spacers. When the authors mention '320 unique sRNA-tRNA interactions', are these with the spacer, the mature tRNA, or both? This point should be clarified.

We only find the reported RyhB-tRNA^{L_{eu}} ETS interaction within ETS or ITS. However, six of the sRNA-tRNA hybrids contain >10nt of pre-tRNA sequence indicating that minimally, these six

interactions are occurring with the pre-tRNA. We have included the line "We note that six sRNA-tRNA interactions contain >10 nt of pre-tRNA sequence indicating that, minimally, these interactions occur before tRNA 5' and 3' maturation" to the Discussion (line 19-21, pg 14).

10) Methods, page14, lines 27-34. To my knowledge, the authors have never published the sequence of the HTF tag. Even if it is just a simple combination of the 6His tag, the optimal TEV sequence and the Flag tag, the authors should specify the exact sequence.

We have now provided the sequence of the HTF tag in Supplementary Table 8 and have uploaded the sequence to NCBI Nucleotide (Accession KX714724).

Referee #3:

In the manuscript "Small RNA interactome of pathogenic E. coli revealed through crosslinking of the degradosome" Waters et al. analyze RNA sequences recovered by UV crosslinking and pulldown of bacterial RNase E. They particularly focus on the small fraction of chimeric sequence reads which are interpreted as derived from bacterial small RNA-mRNA interactions and attempt to validate some of the predicted interactions using reporter assays. Finally they use in vivo assays that a pathogenic E.coli sRNA, Esr41 regulates colicin resistance of this bacterial strain. The approaches used for this study are well-established and thus, the main novelty resides in the characterization of the identified sRNA-mRNA interactions. However, my enthusiasm is dampened by what appears to be a somewhat superficial and not necessarily critical analysis of the high-throughput data in addition to the rather limited set of experiments addressing the biological implications of the predictions. While the results might be of potential interest to the field, the study appears to be a bit premature for publication in a journal of EMBO J's stature.

Specific comments/questions:

1. It is implicitly stated in the text that two replicates of the RNase E CLASH experiments were sequenced. Obviously the raw and the processed data should be deposited in a database, such as GEO.

GEO reviewer links were provided in our cover letter. We have added the accession number for the series (GSE77463) to the Methods section under Preparation of CLASH sequencing libraries.

Perhaps a Supplementary table detailing how many reads per library were obtained, how many mapped, how many of the mapping were non-redundant, etc.

We have now added Supplementary Table 1 that outlines read statistics for our Hfq and RNase E datasets.

Also, since part of the analysis also concerns RNase E binding sites (and not just the chimeras), a list of all binding sites would be helpful for the community as a resource.

We have uploaded the coordinates for RNaseE binding sites with peak height >50 and >100 to GEO under the existing accession number (GSE77463). We have also provided coordinates for RNaseE binding peaks that fall within 1kb of a Hfq binding site (used in our analysis).

From the two replicates, what was the overlap of the RNase E binding sites (not just the chimeras)?

We find that 79.06% (457/578) of peaks identified in the smaller replicate #2 dataset (lower sequencing depth) are also present in replicate #1. We have amended line 33 of pg 5 to "Duplicate UV-crosslinking experiments had a strong correlation in number of reads mapping to transcripts (Spearman's=0.97), and 79.06% of RNase E binding sites in dataset #2 (lower read depth) were also recovered in dataset #1."

Also, the authors claim that 41% of the hybrids found in the datasets were overlapping - it would be good to comment on what this means - is it the background that is overlapping?

We have now estimated background recovery of hybrids and provide p-values for each interaction (detailed for Q3). We would argue that recovery of a hybrid in independent experiments is a reasonably good indicator that it is not background. In line with this, we find that hybrids present in both replicates are generally represented by a higher number of hybrid reads (Figure right, cumulative distribution function of number of hybrids representing each interaction recovered in replicate 1, 2, or both). Ninety-six percent of hybrids present in both datasets have an FDR < 0.05.

2. The separation of signal from noise derived from copurification of abundant cellular RNAs is clearly challenging in all crosslinking/pulldown/sequencing approaches and different methods have been devised (e.g. for eCLIP, iCLIP, PAR-CLIP) to deal with the problem of abortive reverse transcription. In the original paper describing CRAC a characteristic deletion/insertion at the crosslinking site was described - it would be good to orient the reader whether this approach was employed for calling of bona fide binding sites for RNase E.

We have not used deletions to define RNaseE binding sites. We have used the pyCRAC software package to identify statistically significant clusters of reads within our data (pyClusterReads & pyCalculateFDR). We then defined peak maxima within these regions by looking for peaks with a read height > 50 reads and width > 20nt. This information has been added to Supplementary Methods: Analysis of RNase E-RNA binding sites (CRAC data). We had previously demonstrated that deletions were enriched in our Hfq binding data at ARN5m2 sites and unstructured U-U dinucleotides in sRNAs, but were unable to find a clear motif within RNase E binding sites to correlate with reads and deletions.

3. If I read it correctly, it appears that of 21.9 M mapped reads (in both replicates?), ~176k were chimeras and of those 1.7k were consistent with sRNA-mRNA interactions. It is difficult to see how the authors justify focusing on a fraction of reads representing 0.008% of the dataset without very stringent quality control (while virtually ignoring the rest).

We have now included calculations for the probability of random ligation of hybrid halves based on their abundance in the CRAC library. We have based our calculations on those presented by Sharma et al. (Mol Cell 2016) that were devised for background determination in LIGR-Seq data (RNA-RNA interactions captured by psoralen crosslinking in 293T cells). A related study by Lu et al. (Cell 2016), using a similar psoralen crosslinking approach, uses a “connection score” to assess confidence and we have included this calculation for completeness. Notably, the first approach is designed to retain hybrids with a low ratio of hybrid/non-hybrid reads, while the later retains those with a high ratio of hybrids/non-hybrids. This reflects the difficulty in defining background interactions. We have described these calculations in Supplementary Methods: Statistical analysis of hybrids, and provide the calculated FDR and connection score for each hybrid in Supplementary Tables 2, 3 and 8. We had added text directing the reader to the calculations on page 7 lines 23-27.

We agree that our approach yields a large amount of information about RNA structure and RNA-RNA interactions that are bound by RNase E. We have chosen to focus on sRNA-mRNA interactions as there is a need for high throughput tools to profile these regulatory networks. We are currently exploring sRNA interactions with abundant, non-canonical RNA targets recovered in our data but feel this work will require a separate publication.

E.g. in comparison to the Hfq CRAC experiment - how many of those reads were recovered?

We have identified 672 RNase E binding sites that are within 1kb of a Hfq binding site. We find that these Hfq binding sites are intimately associated with the RNase E binding site and have a 5nt 5' offset (pg 7 lines 1-8; Figure 1, and Supplementary Methods: Cumulative plots of non-genomically encoded poly(A) tails, RNase E and Hfq binding sites).

Most importantly, what do the other 175k chimeric reads represent, why are they in such an excess, given that - if I understand it correctly - RNase E does not necessarily interact with dsRNA all by itself but is guided to sRNA-mRNA pairs?

Detailed information on the identity of the ~175K hybrids recovered is provided in Supplementary Table 2. We also present this information graphically in Figure 3 (excluding rRNA interactions). These RNA-RNA interactions represent intra- and inter-molecular interactions in RNAs. As we are only able to capture the site of interaction it is unclear whether these are regulatory fragments of mRNA, tRNA, 5'UTRs etc. (eg: 3'UTR sRNA, tRFs), or interactions between full length transcripts that may function in directing RNA processing analogous to sRNAs. We feel this is an exciting finding and we are currently exploring the functional significance of abundant, non-canonical interactions identified in this study.

How many of these chimeras would one expect by chance (perhaps some analysis in the style of that presented in the Grosswendt et al paper cited would be helpful?)?

As above (top Q3) we have now implemented the background calculation of Sharma et al. Mol Cell 2016.

Also, the authors demonstrate that they recover 14 of 125 predicted sRNA-mRNA pairs (what was the basis for limiting the predicted set to 125
- the database contains >700 interactions)

The 700 interactions in this database are from 53 diverse microorganisms (eg: Agrobacterium, Staphylococcus). We have used 125 sRNA interactions (from 143 that contain some duplicates) reported in E. coli and converted the interaction coordinates to EHEC str. Sakai using BLAST and where required, verified using IntaRNA. We have detailed inconsistencies and errors in sRNATarBase 3.0 in Supplementary Table 5.

- nevertheless, they recover 1700 sRNA-mRNA interactions by themselves, which means that either only ~1% of the possible sRNA-mRNA interactions were previously predicted, or that the CLASH approach calls more interactions than are practicable.

Our results are in agreement with the number of transcripts bound by Hfq in our previous Hfq-CRAC analysis (1253 mRNAs) and by Hfq CLIP-Seq in Salmonella (Holmqvist et al EMBO 2016) where 640 statistically significant Hfq binding sites were recovered. Based on Hfq binding sites alone, it is clear that the 125 sRNA-mRNA interactions currently confirmed in E. coli is almost certainly only a small fraction of those occurring in vivo.

4. I did not notice whether the authors compared their data to RNAseq datasets - what is the number of mRNAs expressed, what is the number of sRNAs expressed, how abundant are they, how many of those are recovered by CLASH, etc.

We have now compared our data to total RNA-Seq (see response to Reviewer 1 Q2).

5. Apparently, most of the sRNA interactions involved other ncRNA molecules. It would be good to comment on the significance of this finding. What is the basis for the interpretation that this is a bona fide interaction and not the sequencing of fragments of the most abundant cellular RNAs? It would be anticipated that these sites would also be found in the Hfq CLASH/CRAC experiments given that Hfq apparently hands over the sRNA/RNA target duplex to RNase E. Does binding of RNase E to these molecules (tRNA/rRNA) result in RNase cleavage?

We are currently working on these non-canonical interactions. As we state in the Discussion for the specific example of tRNA-sRNA interactions, we and others (Lee & Feig 2008; Zhang et al 2003) have demonstrated Hfq interactions with tRNAs. We also observe RNaseE processing and oligoadenylation

of tRNA (data not shown). This is an exciting finding from our dataset and we feel that a careful characterization of these interactions is best presented in a separate manuscript.

6. Figure 3 is left almost uninterpreted and to the imagination of the reader.

Figure 3 is a graphical representation of the RNaseE-CLASH dataset (as noted in Q3 section 3) and provides the reader with the numbers of interactions between each RNA class, number of hybrids representing each interaction, and whether both hybrid halves overlap a Hfq binding site. We discuss Figure 3 on page 9 lines 35-37, pg 10 lines 1-4.

7. Figure 4: in panel B - the seed sequence is identified in ~20% of the putative targets - what other sequence are these targets interacting with. Also, the authors claim that the RyhB sequence GCTCACAT is a seed - even with two mismatches to the CTGGAAGC identified - it would be good to include a discussion about the stringency of the seeds - if in the end 4-5 nt are enough for interaction, it becomes very difficult to predict binding sites.

Thank you for this comment, we agree that the analysis as presented in the original submission was ambiguous. To resolve this, we repeated the analysis using a more stringent set of sRNA and mRNA sequences, and we corrected the thresholds to report only the best match for each motif in the corresponding sRNA sequence. As a result, the ChiX (MicM) motif remains unchanged, but the RyhB motif now covers a single region in the sRNA, which overlaps both regions identified previously. The new motif, AAGCAATG, can be found in more than 50% (260/512) of the mRNA targets. These results can be found in the updated Figure 4 and Supplementary Figure 5. We speculate that most RyhB targets with no similarity to the AAGCAATG motif basepair with other regions of the sRNA, as shown in the heatmap panel of Figure 4.

8. Figure 5: Of the set of 8 high-scoring sRNA-mRNA pairs, it appears that only 4 interactions can be validated (panels A,F,G,H) - the other four either show very little sRNA dependence (B,D,E), or incomplete rescue using compensatory mutations. What is the confidence in the assay? Perhaps it would be good to show some confidence measure for the changes.

We have now added p-values for translational repression in these assays. Complementation of the mutations can be confounded by a number of factors. We discuss the specific example of RyhB-ZapB in the text (pg 11 lines 35-37, pg 12 lines 1-2) where the synonymous mutations within the coding sequence dramatically reduce translation – likely because we introduce a rare leucine codon. We are limited in the number of mutations we can make in this duplex without disrupting the protein sequence. The alternative Glu codon (GAA → GAG) would participate in wobble interactions with RyhB, and the alternative Leu codon (CUU) is similarly rare. Mutation of the Ala codon (GCA) alone would leave 9 consecutive base pairs intact and may not destabilize the duplex.

More generally, point mutations may also disrupt other regulatory sequences or structures (eg: translational enhancing ACA motifs, regulatory secondary structure in the 5'UTR) that cannot be recovered by compensatory mutations in the sRNA, despite bona fide interactions.

We have been collaborating with the lab of Eric Masse to analyze RyeB interactions using their recently published MAPS technique and find that RyeB interacts with RssA – supporting the RyeB-RssA interaction identified in our CLASH data (D. Lalaouna & E. Masse personal comm.). Notably, when assessed using GFP fusions and compensatory base changes, this interaction (Figure 5E) does not dramatically repress RssA, and compensatory base changes do not demonstrate a direct interaction. We feel CLASH and MAPS provide a more direct analysis of sRNA-RNA interactions.

9. Figure 6: Here the genetic interaction of Esr41 (and RyhB) with the chuA, cirA, and bfr genes is analyzed in vivo, and presence of Esr41 represses the genes. Nevertheless, because the authors don't show that this repression is dependent on the direct sequence-specific interaction with Esr41 (e.g. by mutating the binding sites, which the authors have in their hands after all), any of these Esr41 dependent effects could still conceivably be indirect.

This data is presented in Figure 5F-H. We have added a reference to the Figure in the sentence: "Here we have demonstrated that Esr41 regulates expression of the iron transport and storage proteins CirA, ChuA, and Bfr (Figure 5F-H)." (pg 12 line 15-17).

Thank you for submitting a revised version of your manuscript. It has now been seen by all three of the original referees and their comments are shown below. As you will see referees #1 and #2 both find that all major criticisms have been sufficiently addressed and recommend the manuscript for publication, while ref #3 remains somewhat concerned about the low detection rate seen.

Given the overall positive response from the referees I would like to invite you to submit a final revision of your manuscript in which you comment on the remaining concerns from all three referee and also incorporate a brief discussion of the related paper from Melamed et al that was published while your study was in revision.

In addition, I would ask you to address the following editorial issues:

-> We generally require that all information relevant to the main experiments in the manuscript should be included in the Materials and Methods section. I would therefore ask you to move part of the supplemental materials into the main manuscript file. From my side, I would suggest including the section on RNA crosslinking/ligation/sequencing, the confirmation of interactions using sfGFP2 and the Colicin sensitivity and Competitive index experiments. The remaining supplemental methods should be included as part of the Appendix file.

-> As listed in our guide to authors we can accommodate up to 5 typeset EV figures per manuscript and I would therefore ask you to fuse some of the existing 7 EV figures. From my side, I would suggest combining current EV1 and EV2 to one EV figure and turning EV7 into Appendix figure S1. The latter is particularly important since we cannot have main or EV figures be displayed on multiple pages (and I noticed that EV7 is currently contains 8 pages)

-> The legends for the 9 EV tables is currently listed in the main manuscript file but I would ask you to remove them from there and instead include each legend/description as a tab with the individual table files. Please feel free to contact us with any specific formatting questions.

-> For all figures displaying statistics we ask that the number and nature of the replicate (technical vs biological) is included in the accompanying figure legend. I noticed that this information is current missing in figure 5.

REFEREE REPORTS

Referee #1:

This manuscript has been revised, and is much improved in clarity from the original version. I do still think that the low recovery of known sRNA-mRNA interactions is a concern and may limit the general utility of this method, however, it is clear that novel sRNA-mRNA interactions can be identified by this method. Another advantage is the ability to derive information regarding RNase E binding and cleavage sites from this dataset. This will be of broad interest. The authors should add a comment on a paper recently published in Mol Cell by Melamed, et al. That paper utilized a similar strategy, except that Hfq was used to pull down sRNA-mRNA interacting pairs. The authors should comment on the relative benefits and caveats of their method compared with the method described in the other manuscript.

Referee #2:

The authors have made a comprehensive revision of the manuscript including new experimental data as well as a more detailed statistical analysis of some of the results. They have also made a comprehensive rebuttal to all the concerns raised by the reviewers. From the point of view of RNA

biology in bacteria, the revised manuscript is much more than the description of a novel sequencing method. With respect to the CLASH results, it is not surprising that the capture of hybrids by ligation is inefficient. Although not fully compelling due to the low number of reads, the authors have done a good job providing valid arguments against the possibility that hybrids are the results of low-level non-specific ligation events.

Minor comments.

In Figure 2F, why is tRNA separated into two different sectors? Is one sector supposed to be tmRNA? Page 10, lines 1 and 2. What do the authors mean by 'major target for select sRNA.' Page 12, line 11. Figure 3C. Figure 3 does not have sub-panels.

Referee #3:

The manuscript "Small RNA interactome of pathogenic *E. coli* revealed through crosslinking of RNase E" describes the analysis of bacterial small RNA-target interactions using high-throughput sequencing. The main part of the analysis concerns so-called chimeric sequence reads generated by the authors' CLASH method which is able to recover direct interactions between small RNAs and their targets. A number of newly discovered small RNA-target interactions were then validated using reporter assays.

The revised manuscript has now incorporated a number of additional analyses and Supplementary Tables summarizing the data in reaction to the reviewer's comments. Not much new experimental data was added to increase the confidence in the method and conclusions.

While the new analyses clearly strengthen the presentation of the CLASH data, the lack of new data results in the persistence of my main concerns:

This work is concerned with the systems-wide characterization of RNase E and the CLASH approach is ideally suited to help identify sRNA-guided RNase E targeting, which is preceded by sRNA-Hfq complex binding to targets. Nevertheless, the chimeric reads the authors focus on represent 1700 out of 11 M sequence reads (in their second replicate 194 out of 2.1 M reads). These are minuscule numbers and the confidence in the results is further undermined by the fact that only 60% of their chimeras pass the FDR criterion. The additional ~170k chimeric reads remain ignored and little is offered as interpretation as to why they are ~100 times more abundant than sRNA-mRNA interactions and what their importance is. The low recovery of chimeric reads from sRNA-mRNA pairs is particularly disappointing in the light of the recent publication of a conceptually similar paper in *Mol. Cell* by Margalit et al. that presents analysis of sRNA-mRNA chimeras from Hfq crosslinking. The publication demonstrates that it is possible to recover a sizable number of sRNA-mRNA interactions (~200 k out of 1.4 M chimeras). In addition Margalit et al. offer insights into the dynamics of sRNA-mRNA interactions moving the analysis beyond correlative observations, as well as experimental validation of their crosslinking approach by analyzing the interactome of bacteria missing the seed sequence in one of their sRNAs. Thus, the study by Margalit et al. may serve as the paradigm of what could be extracted from a technically adequate CLASH experiment.

Another (in my opinion unfortunate) result of the focus on the 1700 reads is that the relationship between Hfq and RNase E is reduced to a single panel (Fig 1E). The available data from the RNase E and Hfq CRAC reads would allow a deeper exploration in order to build hypotheses on the relationship of these important posttranscriptional regulatory proteins in bacteria.

In summary I cannot judge whether the study is sufficiently developed for publication in a journal with the broad audience of EMBO J, or whether it would be better suited for a more specialized journal.

We thank the referees for their thoughtful comments. The MS has been revised as requested and we feel that this has resulted in a significant improvement.

Referee #1:

This manuscript has been revised, and is much improved in clarity from the original version. I do still think that the low recovery of known sRNA-mRNA interactions is a concern and may limit the general utility of this method, however, it is clear that novel sRNA-mRNA interactions can be identified by this method. Another advantage is the ability to derive information regarding RNase E binding and cleavage sites from this dataset. This will be of broad interest. The authors should add a comment on a paper recently published in Mol Cell by Melamed, et al. That paper utilized a similar strategy, except that Hfq was used to pull down sRNA-mRNA interacting pairs. The authors should comment on the relative benefits and caveats of their method compared with the method described in the other manuscript.

We agree that a short comment in the discussion is warranted. We have added to following paragraph:

“While this work was in revision, a related technique for sequencing sRNA-RNA interactions termed RIL-Seq was described (Melamed *et al*, 2016). This is conceptually similar to RNase E-CLASH except that Hfq is used as a scaffold to capture sRNA-RNA duplexes and the purification is performed under native conditions as opposed to CLASH that uses a stringent purification protocol. Stringency is introduced into RIL-Seq analysis *in silico* where hybrid reads are filtered for statistical enrichment. We find a comparable number of statically significant sRNA-mRNA interactions are recovered by both techniques in log phase cells (633 using RIL-Seq and 782 using RNase E-CLASH) and similar sRNA seed regions and motifs are recovered for abundant sRNAs (eg: ArcZ, MgrR, GcvB, and CyaR) suggesting that both techniques capture *bona fide* sRNA-RNA interactions. Notably, the pools of RNA-RNA interactions recovered in association with Hfq and RNase E are expected to be different. RNase E processes a broad range of RNA species and is expected to associate with a subset of all sRNA-mRNA interactions that specifically result in target degradation.”

We have prepared a more detailed comparison of RIL-Seq and RNase E-CLASH for Reviewer #3 (below).

Referee #2:

The authors have made a comprehensive revision of the manuscript including new experimental data as well as a more detailed statistical analysis of some of the results. They have also made a comprehensive rebuttal to all the concerns raised by the reviewers. From the point of view of RNA biology in bacteria, the revised manuscript is much more than the description of a novel sequencing method. With respect to the CLASH results, it is not surprising that the capture of hybrids by ligation is inefficient. Although not fully compelling due to the low number of reads, the authors have done a good job providing valid arguments against the possibility that hybrids are the results of low-level non-specific ligation events.

Minor comments.

In Figure 2F, why is tRNA separated into two different sectors? Is one sector supposed to be tmRNA?

Thank you for bringing this to our attention – the 5.9% wedge should be labeled “other”. We have amended the figure.

Page 10, lines 1 and 2. What do the authors means by 'major target for select sRNA.'

We find that not all sRNAs interact with tRNA in our dataset. Conversely there a small number of sRNAs that appear to have abundant interactions with tRNA.

We have amended the text to “major target for a subset of sRNAs.”

Page 12, line 11. Figure 3C. Figure 3 does not have sub-panels.

Amended

Referee #3:

The manuscript "Small RNA interactome of pathogenic *E. coli* revealed through crosslinking of RNase E" describes the analysis of bacterial small RNA-target interactions using high-throughput sequencing. The main part of the analysis concerns so-called chimeric sequence reads generated by the authors' CLASH method which is able to recover direct interactions between small RNAs and their targets. A number of newly discovered small RNA-target interactions were then validated using reporter assays.

The revised manuscript has now incorporated a number of additional analyses and Supplementary Tables summarizing the data in reaction to the reviewer's comments. Not much new experimental data was added to increase the confidence in the method and conclusions.

While the new analyses clearly strengthen the presentation of the CLASH data, the lack of new data results in the persistence of my main concerns:

This work is concerned with the systems-wide characterization of RNase E and the CLASH approach is ideally suited to help identify sRNA-guided RNase E targeting, which is preceded by sRNA-Hfq complex binding to targets. Nevertheless, the chimeric reads the authors focus on represent 1700 out of 11 M sequence reads (in their second replicate 194 out of 2.1 M reads).

The reviewer has pointed out an important point regarding terminology. 2714 unique hybrid reads representing 1733 unique interactions were recovered. Unique interactions potentially represent multiple PCR duplicates, that are collapsed into unique hybrid reads, that are further collapsed into unique interactions. We have amended the terminology to distinguish between reads, hybrids, and interactions in Table EV1.

These are minuscule numbers and the confidence in the results is further undermined by the fact that only 60% of their chimeras pass the FDR criterion.

Please see our comparison of RIL-Seq and CLASH data in Table 1 and analysis of raw data below. Comparable numbers of FDR corrected interactions are recovered.

The additional ~170k chimeric reads remain ignored and little is offered as interpretation as to why they are ~100 times more abundant than sRNA-mRNA interactions and what their importance is. The low recovery of chimeric reads from sRNA-mRNA pairs is particularly disappointing in the light of the recent publication of a conceptually similar paper in *Mol. Cell* by Margalit et al. that presents analysis of sRNA-mRNA chimeras from Hfq crosslinking. The publication demonstrates that it is possible to recover a sizable number of sRNA-mRNA interactions (~200 k out of 1.4 M chimeras).

Please see analysis of the raw data from Melamed *et al* below. It is possible that the number of interactions reported has been increased by PCR duplicates.

In addition Margalit et al. offer insights into the dynamics of sRNA-mRNA interactions moving the analysis beyond correlative observations, as well as experimental validation of their crosslinking approach by analyzing the interactome of bacteria missing the seed sequence in one of their sRNAs. Thus, the study by Margalit et al. may serve as the paradigm of what could be extracted from a technically adequate CLASH experiment.

Another (in my opinion unfortunate) result of the focus on the 1700 reads is that the relationship between Hfq and RNase E is reduced to a single panel (Fig 1E). The available data from the RNase E and Hfq CRAC reads would allow a deeper exploration in order to build hypotheses on the relationship of these important posttranscriptional regulatory proteins in bacteria.

In summary I cannot judge whether the study is sufficiently developed for publication in a journal with the broad audience of EMBO J, or whether it would be better suited for a more specialized journal.

The major difference between our approach and that described by the Maragalit lab (RIL-Seq) is the stringency of the purification (as stated in their manuscript). RIL-Seq data is generated from UV-crosslinking FLAG tagged Hfq under native conditions, whereas RNaseE-CLASH uses a highly stringent dual FLAG-His purification ie: native purification over anti-FLAG resin and then stringent purification in 6M guanidinium using the 6xHis tag.

For RIL-Seq, stringency is introduced *in silico* where hybrid reads are filtered for enrichment using an Odds Ratio, and by removing interactions that are represented by <10 hybrid reads.

We have compared the results for each approach in attached Table 1.

RNase E-CLASH recovers comparable sRNA-mRNA interactions in less reads

Using native purification conditions in log phase (as used in our experiments), RIL-Seq recovers 633 significant sRNA-mRNA interactions (S-chimeras; 918 total RNA-RNA interactions; Tables S1 and S2 in Melamed *et al.*) from 6 libraries representing 58M reads. The average recovery of non-redundant sRNA-mRNA hybrids in log phase cells for RIL-Seq is 10.87 interactions/M reads (max. S-chimeras from single library is 45), and in our replicate RNase E-CLASH we recover 36.71 interactions/M reads (max. from a single library is 96).

The RIL-Seq analysis benefited from the large number of sequencing libraries prepared, as 257 of the 633 sRNA-mRNA interactions in log phase cells only meet the statistical filter (>10 reads, FDR<0.05) when the data from the 6 replicate experiments are pooled together (58M reads). Similarly, from the 12 sequencing libraries reported in the manuscript (representing 135.3 M reads), 1631 sRNA-mRNA interactions are recovered (sRNA with 5UTR or CDS or 3UTR in Table S1). Of these, 438 (26.8%) only reach statistical significance when the data from all 12 libraries (135M reads) are pooled.

For future analyses, the number of targets that are identified by CLASH could also be increased by greatly expanding the number of samples, but this would entail very considerable additional sequencing costs and time.

RIL-Seq does not appear to remove PCR artifacts

The bioinformatic pipeline used to prepare the RIL-Seq data does not appear to include steps to eliminate PCR duplicates in the data. Our RNase E-CLASH library preparation includes incorporation of a 3-nt random sequence into the 5' linker to allow us to identify PCR duplicates and collapse these reads into a single sequence. All of our analysis represents PCR collapsed data. The final step in preparation of both RIL-Seq and RNase E-CLASH libraries is the PCR amplification of the cDNA library and we routinely find that removing PCR duplicates (reads with the same sequence) reduces the size of our libraries up to 10-fold. The number of unique hybrid reads representing an RNA-RNA interaction can be significantly inflated in the uncollapsed data and will impact on the statistical significance of the interactions identified as the majority of sRNA-mRNA interactions identified by RIL-Seq are represented by <100 reads (log phase [68.25%], stationary phase [75.96%], iron limitation [69.57%]).

We have done a preliminary analysis of library FLAG101 (Hfq-Flag_Log_Phase_CL_Repeat_1). Melamed *et al* report 12.4M quality filtered reads and 6.2M single mapping reads (concordant [proper] pairs within the same transcript or within 1000nt [Melamed *et al.* Table S1]). We have quality filtered the sequencing reads (ERR1547242_1.fastq, ERR1547242_2.fastq; 20.57M paired reads) using SeqPrep, yielding 16.6M reads. Removing PCR duplicates from these 16.6M reads using pyFastqRemoveDuplicates from the pyCRAC software package (Webb *et al* Genome Biology 2014) reduces the dataset to 4.6M suggesting that 62% of the data is PCR duplicates. Similar results are obtained by filtering the raw sequencing data with FastUniq (Xu *et al* PLoS One 2012) yielding 5.47M reads.

Mapping these 4.6M reads using novoalign yields 733K proper-pairs (896K proper-pairs using bowtie2 read aligner) where the proper-pairs map unambiguously within 1000nt of each other (cf. 6.2M pairs reported; ie: 8.45-fold reduction in sequence depth). The authors have used different software for processing their data (that will affect adapter trimming, quality filtering, mapping etc.) but we feel that PCR amplification artifacts may have increased the read statistics.

Hfq is not a good “scaffold” under stringent conditions

We had earlier performed CLASH on Hfq using our stringent purification protocol and found that a very limited number of hybrid reads were recovered (presented in Table EV1). From a single Hfq-CLASH library of 63M reads we were able to recover 221 sRNA-mRNA interactions. Using RNase E-CLASH we recover 1733 sRNA-mRNA interactions from 21.8M reads (2 libraries). 782 of these interactions have an FDR of <0.05, and 389 are represented by 2 or more unique hybrid reads (ie: PCR duplicates are removed from the library and the hybrid is represented by 2 or more non-PCR duplicates). Notably, we were able to show that two sRNA-mRNA interactions represented by a single hybrid read recovered under our stringent purification conditions were functional.

Both RIL-Seq and RNase E-CLASH advance sRNA research

While RIL-Seq is clearly a useful tool to identify sRNA-mRNA interactions, there are advantages to using stringent purification conditions, particularly when exploring unusual or unexpected RNA-RNA interactions. We found that hybrid reads were not well recovered on Hfq under stringent purification conditions. RNase E appears to interact more stably with sRNA-RNA pairs allowing stringent purification and recovers comparable numbers of statistically significant sRNA-mRNA interactions when compared to native purification on Hfq.

It should be noted that the pool of RNA-RNA interactions recovered is expected to be different between Hfq and RNase E. We anticipate that RNase E will be associated with the subset of all sRNA-mRNA interactions that specifically result in target degradation. So each technique is sampling from different pools on RNA-RNA interactions.

While RIL-Seq and RNase E-CLASH differ in approach and growth conditions (that affect sRNA-mRNA recovery) we note that many of the sRNA seed motifs identified are very similar (ArcZ, MgrR) or at the same site (GcvB, CyaR, MicA [5nt away]).

Overall, both RIL-seq and CLASH are able to identify sRNA-mRNA interactions *in vivo* and each has advantages and disadvantages.

Table 1. Comparison of RIL-Seq and RNase E-CLASH recovery of RNA-RNA interactions (RIL-Seq data from Table S1 and 2 in Melamed *et al*)

Approach	Strain	Condition	Reads (M)	Libraries	Total hybrid reads*	Total RNA interactions	sRNA-mRNA hybrid reads	sRNA-mRNA interactions	sRNA-mRNA interactions with $p < 0.05^{**}$	sRNA-mRNA interactions/M reads (max from single library)
RIL-Seq	MG1655	All conditions	135.3	12	1.178M	2817	818K	-	1631	12.05 (max S-chimeras 93)
RIL-Seq	MG1655	Log phase (rich media)	58.2	6	274K	1027	209K	-	633	10.87 (max. S-chimeras 45)
RNase E-CLASH	EHEC str. Sakai	Log phase (MEM-HEPES media)	21.3	2	349K	176K	2714	1733	782	36.71 (max. sRNA-mRNA 96)

*S-chimeras for RIL-Seq and total hybrids for RNase E-CLASH

**RIL-Seq determines FDR using a Fishers exact test and >10 reads, RNase E-CLASH uses the probabilistic approach described by Sharma *et al* Mol cell 2016.

3rd Editorial Decision

11 October 2016

Thank you for submitting this final revision of your manuscript to The EMBO Journal. I have now gone through your response to the remaining minor referee concerns and I am pleased to inform you that the study has been accepted for publication here.

Corresponding Author Name: Jai Tree

Manuscript Number: EMBOJ-2016-94639